# TfR1 facilitates influenza virus endocytosis and uncoating by interacting with NA and M1 via extracellular and intracellular domains

Xinchen Wang[1,2,3,4☯], Yuanhao Li[1,2,3,4☯], Dezhong Ji[1,3,4☯*], Yiming Wang[1,2,3☯], Xiaoyang Wang[1,2,3], Kangming Guo[1,2,3], Mengyang Wang[1,2,3], Yu Mu[2], Chen Qin[3], Tao Yuan[1,2,3], Yuanjie Zhang[1,2,3], Zhiqian Chen[1,2,3], Xingxing Zhu[1,2,3], Xiaohui Zhang[1], Honghui Jiang[3], Qiuchen He[1,2], Chuanling Zhang[1,4], Sulong Xiao[1,3*], Lihe Zhang[1,3], Demin Zhou [1,2,3,4*]

**1** State Key Laboratory of Natural and Biomimetic Drugs, School of Pharmaceutical Sciences, Peking University, Beijing, China, **2** Shenzhen Bay Laboratory, Gaoke International Innovation Center, Shenzhen, China, **3** Peking University Ningbo Institute of Marine Medicine, Ningbo, China, **4** Peking University-Yunnan Baiiyao International Medical Research Center, Beijing, China

☯ These authors contributed equally to this work.
* jidezhong@bjmu.edu.cn (DJ); slxiao@bjmu.edu.cn (SX); deminzhou@bjmu.edu.cn (DZ)

## Abstract

An intriguing enigma in virology is the utilization of transferrin receptor 1 (TfR1) by various viruses as an entry portal into host cells, a mechanism that remains relatively underexplored. In this study, we report a strategy to investigate the multifaceted aspects of viral entry, using Influenza A viruses (IAVs) as a model system. By decorating the sialylated viral envelope with photo-crosslinking moieties, we identify and elucidate the pivotal role of TfR1 in this process. Our results demonstrate that TfR1 initially functions as a receptor, interacting with the viral neuraminidase (NA) through its extracellular apical domain, thereby initiating viral endocytosis. Subsequently, TfR1 acts as a matrix degradator, engaging its intracellular stop-transfer sequence with the viral matrix protein 1 (M1), which in turn triggers proteasome- and aggresome-mediated nucleocapsid uncoating. The identification of the molecular interactions between TfR1 and NA, as well as the reciprocal degradation of TfR1 and M1 not only illuminates a cellular pathway that enriches our understanding of viral entry mechanisms but also presents exciting avenues for the development of innovative antiviral strategies beyond IAVs.

## Author summary

Influenza viruses are masters of exploiting human cells, but how they hijack cellular machinery to enter and infect hosts remains partially understood. In this study, we combined cutting-edge metabolic glycan labeling and photo-crosslinking technologies to uncover a dual role of the iron transporter transferrin receptor 1 (TfR1) in influenza A virus (IAV) infection. Using innovative chemical

**Data availability statement:** All relevant data are within the manuscript and its Supporting Information files.

**Funding:** This work was supported by the National Natural Science Foundation of China (grant no. 82130100 to DZ, grant no. 82473830 to DJ, grant no. 82204258 to DJ, grant no. 823B1004 to TY), National Key R&D of Program of China (grant no. 2024YFA0917500 to DZ), The Ningbo Key Science and Technology Development Program (grant no. 2022Z136 to DZ), The Ningbo Yongjiang Talent Program (grant no. 2024A-167-G to HJ). The funders had no role in study design, data collection and analysis, decision to publish, or preparation of the manuscript.

**Competing interests:** The authors have declared that no competing interests exist.

tools to tag viral surface sugars, we discovered that TfR1 acts as a "gateway" for the virus: first, it binds to the viral neuraminidase (NA) protein to trigger entry into cells via extracellular domains, and later, it degrades virus structural matrix protein (M1) via intracellular domains. This two-step mechanism bridges critical gaps in our understanding of how viruses enter cells and uncoat their genetic material. Importantly, TfR1's ability to degrade viral proteins extends to other pathogens like measles and rabies viruses, suggesting a universal strategy for antiviral therapies. By revealing how a single host protein facilitates both viral invasion and self-destruction, our work opens new avenues for designing drugs that disrupt these interactions, potentially offering broad protection against multiple deadly viruses. This discovery not only deepens our knowledge of viral infections but also highlights TfR1 as a promising target for next-generation antiviral treatments.

## 1. Introduction

RNA viruses intricately adorn their envelopes with diverse glycans [1,2] and possess a remarkable ability to exploit host cells, leading to severe diseases and even pandemic outbreaks [3,4]. The successful entry of RNA viruses into host cells relies on a multifaceted, highly choreographed, multi-step process [5,6]. Exemplified by Influenza A viruses (IAVs), this process involves several critical stages: initial viral attachment to host cells [7], penetration of the host cell membrane, fusion between viral and endosomal membranes [8], uncoating of the viral matrix to release viral RNAs into the cytoplasm [9], and eventual nuclear entry for replication [10]. IAVs possess glycosylated envelope proteins, hemagglutinin (HA), and neuraminidase (NA), which facilitate interactions with host cells [11–13]. It is widely accepted that HA recognizes host sialic acid receptors during the initial viral entry [14,15], while NA plays a crucial role in the release of newly formed viruses from the host cell plasma membrane during budding [16,17]. Despite the ubiquitous presence of terminal sialic acids on various human cells [18], only specific tissues and organs are particularly vulnerable to IAV infection. This raises an intriguing question about the underlying mechanisms of tissue specificity.

Understanding the intricate virus-host interactions is fundamental for elucidating how viruses exploit cellular machinery for entry but remains a significant challenge [19]. Several host proteins have been proposed as potential entry factors for IAVs, including the epithelial growth factor receptor (EGFR) [20], free fatty acid receptor 2 (FFAR2) [21], nucleolin [22], and the voltage-dependent calcium channel Cav1.2 [23]. However, the continued viral entry upon knockdown or knockout of these proteins implies a redundancy in entry mechanisms and indicates the existence of other, yet unidentified, entry factors [24]. A recent study revealed that the binding of IAV virions to TfR1 via sialic acids triggers viral entry, with IAVs exploiting TfR1 recycling as a revolving door mechanism to enter host cells [24]. However, this finding seems likely argues against previous observation that sialic acid is not a requisite for effective IAV

infection in GnT1-deficient CHO 15B and HEK 293S cells [25], suggesting that while sialic acid may facilitate initial viral attachment to the cell membrane, it does not always guarantee efficient entry [26,27]. Critical questions remain, including the identification of protein receptors responsible for anchoring IAVs to host cells and initiating endocytosis [28].

In this study, we present a strategy that involves decorating IAVs with photo-crosslinking moieties on sialylated envelope components, enabling us to elucidate the sequential process of IAV entry with a particular focus on host factors involved. Our findings reveal that TfR1 emerges as a crucial factor, not only initiating IAV endocytosis by interacting with the viral envelope NA through its extracellular apical domain but also playing a critical role in uncoating viral ribonucleo-proteins (vRNPs). This uncoating process involves the interaction of TfR1's intracellular stop-transfer sequence with the viral matrix M1 protein, converting M1-TfR1 complex into a misfolded protein. This mechanism unveils the potential factor responsible for triggering matrix protein 1 (M1) degradation [29], potentially involving proteasome and HDAC6 mediated pathways [30–32]-a pivotal step following HA/endosomal fusion to release viral RNAs into the cytoplasm. Our discoveries not only fill critical gaps in the understanding of the viral entry pathway but also pave the way for the development of novel antiviral strategies beyond IAVs.

## 2. Results

### 2.1. Decorate IAVs with probes for photo-crosslinking host proteins involved in entry

Our investigation commenced with the metabolic glycan labeling of the WSN strain (H1N1 subtype), serving as a representative model. This involved introducing azido groups through the transfection of IAV plasmids constructs into 293T cells in the presence of $Ac_4ManNAz$—a monosaccharide featuring a D-N-acetylmannosamine core and an azido-modified N-acyl side chain. This process aimed to generate IAV particles, potentially featuring modified terminal sialic acids, denoted as $IAV-N_3$ (Figs 1A and S1A). These modified virus particles then underwent conjugation with DIBO-488, a green fluorescence ligand containing a DIBO moiety suitable for click reaction (Fig 1B), or DIBO-Biotin, resulting in the formation of IAV-Biotin particles, as confirmed by immunofluorescent staining and western blotting analysis of infected A549 cells (Fig 1C). This observation indicates the metabolic conversion of $Ac_4ManNAz$ into terminal sialic acid on the sialylated envelope of the WSN strain, especially HA and NA, significantly contrast to the absence of $Ac_4ManNAz$ as a negative control. Afterwards, a probe (DIBO-DAZ-Biotin) featuring three essential components—DIBO, diazirine (DAZ), and biotin—attached to a lysine skeleton, was introduced using the same method. Importantly, this metabolic glycan labeling of the WSN virus did not affect the virus's infectivity, as confirmed through the infectivity assay by comparison with the wild-type WSN virus (S1B Fig).

A549 cells were then infected with IAV-DAZ-Biotin particles (MOI = 50) at 4°C for 2 hours to facilitate virus attachment, followed by a shift to 37°C for 5 minutes. At this stage, UV irradiation or not (a negative control) was applied for 20 minutes, and crosslinked proteins were subsequently enriched through streptavidin agarose, separated using 8% SDS-PAGE detected by anti-HA, -NA or -TfR1 antibody (Figs 1E and S1A). Concurrently, a parallel experiment involved pre-incubating A549 cells with $Ac_4ManNAz$ for 12 hours before infection with wild-type WSN virus, followed by identical UV irradiation conditions as the background. Utilizing mass spectrometry (UPLC-MS) analysis, a total of 195 proteins were identified across three independent photo-crosslinking experiments. Subsequent volcano plot analysis pinpointed 10 proteins as potential factors implicated in IAV entry, including transferrin receptor 1 (TfR1) (Fig 1D), identified as a putative receptor involved in the entry processes of several specific viruses [24,33,34], and other attractive targets EGFR, LDLR, and ITGA3.

### 2.2. Identify TfR1 as an IAV entry mediator critically for triggering viral endocytosis

Subsequently, TfR1 knockdown experiments were conducted in A549 cells, revealing a substantial reduction in WSN infection (MOI = 0.5), as evidenced by both western blotting and immunofluorescence assays (Fig 2A and 2B). This

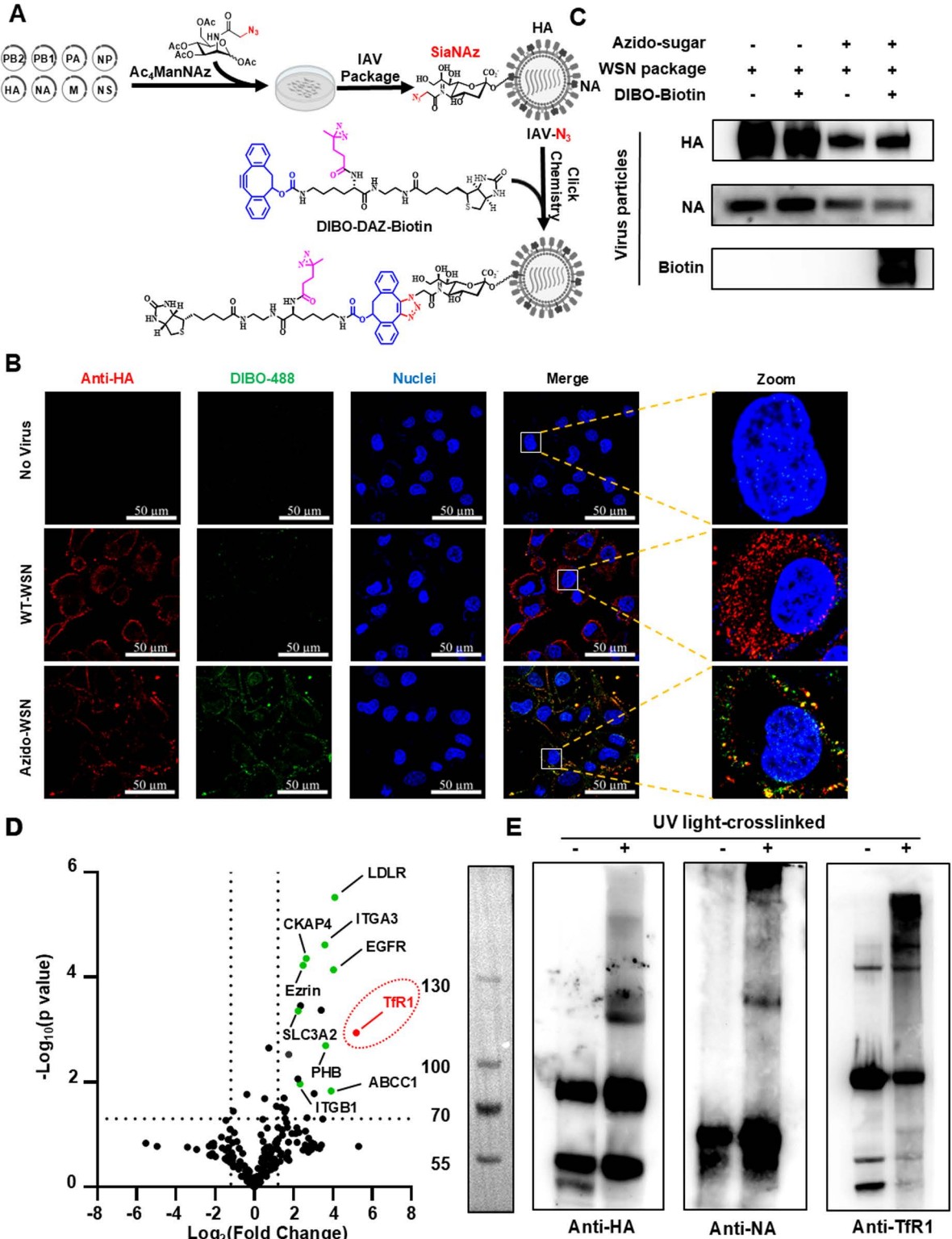

**Fig 1. Decoration of the sialylated envelope of IAVs with probes for photo-crosslinking host proteins involved in virus entry. (A)** Schematic representative of metabolic conversion of Ac$_4$ManNAz into terminal sialic acid on sialylated envelope for generation of azido-displayed IAV (IAV-N$_3$), followed by click conjugation with DIBO-DAZ-Biotin for labeling IAVs with photo-crosslinking probe. Schematic created using BioRender (https://Biorender.

com) **(B)** Immunofluorescent staining depicting A549 cells infected with IAV or azido-displayed IAV particles (MOI = 50) pre-treated with DIBO-488 for 1h on ice. HA (green), Nuclei (DAPI, blue). Scale bars represent 50 μm. **(C)** Western blotting characterization of purified IAV particles generated in the presence or absence of Ac$_4$ManNAz, subsequently conjugated with DIBO-Biotin. Detection using HA/NA/Biotin antibodies. **(D)** Presentation of various crosslinked candidate proteins, with TfR1 specifically highlighted in red, employing a volcano plot. Proteins showing a ratio of 2 or higher, relative to the control group, were considered for subsequent analysis. This dataset encompasses information derived from three independent experiments. **(E)** Analysis of photo-crosslinked products from A549 cells infected with IAV (MOI = 50) pre-labeled with DIBO-DAZ-Biotin or left untreated. Cells were incubated with virus for 1 h on ice, then shifted to 37°C for 5 min. Samples were either exposed to UV irradiation (+) or not (−) for 20 min. Crosslinked proteins were enriched using streptavidin agarose, separated by 8% SDS-PAGE, and detected by western blotting with anti-HA, anti-NA, or anti-TfR1 antibodies. The observed mobility shift of TfR1 in the UV-treated group indicates successful photo-crosslinking between TfR1 and viral components.

observation was further supported by assessing virus infections of A549 cells with various IAV strains (A/WSN/1933 H1N1, A/PR/8/1934 H1N1, and A/Aichi/2/1968 H3N2), underscoring the critical role of TfR1 across diverse IAV strains (Figs 2C and S2F). Notably, a discernible correlation between cell lines' susceptibility to IAV infection and their TfR1 expression levels was observed: cell lines expressing higher levels of TfR1, such as MDCK, A549, and U87 cells, exhibited heightened infectivity, while 293T cells with lower TfR1 expression displayed reduced susceptibility, and CHO cells lacking TfR1 expression demonstrated minimal M1 upon IAV infection (Figs 2D and S1C). Confocal immunofluorescence imaging further supported TfR1's role in IAV entry, showcasing colocalization of TfR1 (green) with viral HA or NA proteins (red) during IAV infection, resulting in overlapping yellow staining (S1D Fig). We then compared sialic acid versus TfR1 as potential receptors in mediating IAV entry. Notably, pretreatments of A549 cells with neuraminidase resulted in a more substantial decrease (~95%) in virus infection, significant contrast to the pretreatment of IAV virus with neuraminidase that led to ~50% infectivity decrease (Figs 2E and S2A) This finding was reinforced by experiments wherein TfR1 overexpression in neuraminidase pre-treated A549 cells moderately increased virus infection, though significantly lower than that of untreated cells (Figs 2E and S2A). This finding was also strengthened by experiments wherein TfR1 overexpression in Lec1 cells (CHO cells deficient in N-acetylglucosaminyltransferase I (GnT-I), an enzyme critical for normal N-glycosylation.) moderately increased virus infection (S2E Fig). Clearly, both sialic acid (mainly from host cells rather than viruses) and TfR1 are critical mediators involved in IAV entry.

The subsequent focus of our investigation was to scrutinize TfR1's involvement in virus entry, either attachment or endocytosis. To assess the role of TfR1 in viral attachment, A549 cells and TfR1-knockdown A549 cells were incubated with IAVs (MOI = 10, with culture medium containing 1% FBS to exclude unspecific binding) at 4°C for one hour. After washing three times, we quantified the attached viruses using RT-qPCR and found a ~25% reduction in virus attachment in TfR1-knockdown A549 cells compared to those treated with negative siRNA, highlighting the significant role of TfR1 in the initial attachment of IAV particles to host cells. To evaluate TfR1's role in the endocytosis of attached virus particles, we transferred the aforementioned cells from 4°C to 37°C and incubated them for an additional two hours with or without NaN$_3$, a viral endocytosis inhibitor. RT-qPCR analysis revealed that TfR1-knockdown cells exhibited a ~70% reduction in the number of internalized viruses compared to control cells (Figs 2F and S2G). The viral internalization was further investigated by exposure of A549 cells, TfR1-knockout A549 cells, and CHO cells to sulfo-NHS-SS-Biotin-labeled IAVs [35] (MOI = 5) under identical conditions. Flow cytometry analysis, using AF488-conjugated streptavidin [36], showed an ~80% reduction in mean fluorescence intensity in TfR1-knockout cells compared to parental A549 cells (S2B Fig). This reduction was comparable to that observed in CHO cells, which inherently lack TfR1 expression. Moreover, overexpression of TfR1 in TfR1-knockout A549 cells restored IAV endocytosis (Figs 2G, S2B and S2G). Thus, it seems likely that TfR1 is more crucial for IAV endocytosis than for initial attachment to the cell surface, despite they are not independent steps.

Given TfR1's identification as a primary component in the canonical clathrin-mediated endocytosis (CME) pathway for transferrin (Tf) endocytosis [36,37], we investigated whether IAVs utilize this pathway for viral endocytosis through siRNA knockdown experiments. Significant inhibition of IAV infection was observed upon knockdown of clathrin and transferrin receptor trafficking protein (TTP) in A549 cells, mirroring the effect observed with TfR1 knockdown (S2C Fig). Confocal

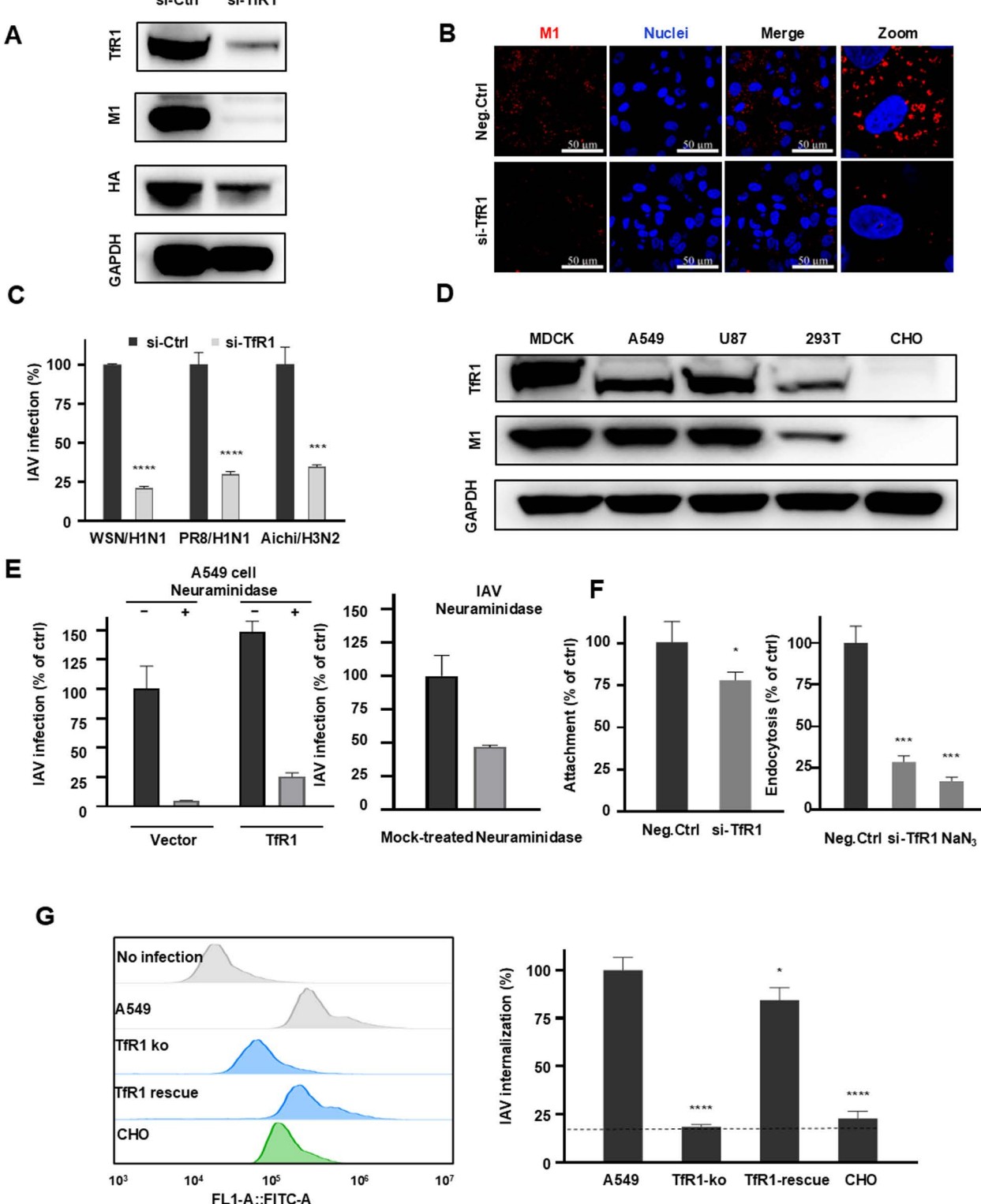

**Fig 2. Elucidation of host TfR1 as an IAV entry mediator critically for triggering viral endocytosis. (A)** Western blot analysis examining the impact of TfR1 knockdown in A549 cells on IAV (A/WSN/1933 (H1N1)) 12 hours infection (MOI = 0.5). Equal loading was confirmed by detecting GAPDH. **(B)** Immunofluorescent staining of A549 cells and cells with TfR1 knockdown 4h post -infection (A/WSN/1933 (H1N1)) (MOI = 0.5). **(C)** Evaluation of TfR1

knockdown's inhibitory effect on IAV infection (MOI = 0.5) in A549 cells. Viral RNA levels were quantified by RT-qPCR at 12 hours post-infection across multiple strains: A/WSN/1933 (H1N1), A/PR/8/1934 (H1N1), and A/Aichi/2/1968 (H3N2). **(D)** Western blot characterization investigates the correlation between TfR1 expression levels in different cell lines and their susceptibility to IAV infection (MOI = 0.5). Equal loading was confirmed by detecting GAPDH. **(E)** Comparative evaluation of sialic acid versus TfR1 as potential receptors in mediating IAV entry. A549 cells transfected with TfR1/vector plasmid were treated with neuraminidase (50 units/mL) or BSA at 37°C for 2 hours before infection with IAV (A/WSN/1933 (H1N1), MOI = 0.5) (*left panel*). Alternatively, IAV was pretreated with neuraminidase or BSA at 37°C for 2 hours before infecting A549 cells (*right panel*). After 12 hours, infected cells were lysed and quantified by RT-qPCR. **(F)** Comparative evaluation of TfR1 as a potential receptor mediating IAV attachment and endocytosis. A549 and TfR1-knockdown cells were incubated with either unlabeled IAVs (A/WSN/1933 (H1N1), MOI = 10) or sulfo-NHS-SS-Biotin-labeled IAVs (MOI = 5) at 4°C for 1 hour (to allow attachment) or subsequently shifted to 37°C for 2 hours (to allow endocytosis). After washing, attached viruses were quantified by RT-qPCR, and internalized viruses were measured by flow cytometry. **(G)** Flow cytometry analysis assessing TfR1 involvement in IAV entry, quantifying internalized biotin-SS-labeled IAVs (A/WSN/1933 (H1N1)) (MOI = 5) in A549 cells, TfR1-knockout A549 cells, and CHO cells. TCEP treatment was applied to cleave disulfide bond and remove the biotin tag on non-internalized IAV particles, followed by fixation, permeabilization and incubation with AF488-conjugated streptavidin. Unpaired t-tests were used for statistical analysis with corresponding p-values. *p < 0.05; **p < 0.01; ***p < 0.001; and ****p < 0.0001, while "n.s." indicates non-significance.

immunofluorescence imaging of IAV-infected A549 cells confirmed the colocalization of IAV (blue), TTP/CLTA (Clathrin light chain A) (red), and TfR1 (green), as indicated by the white spots demonstrating overlapping staining (S2D Fig). Consistent with these findings, the clathrin pathway inhibitors Chlorpromazine (CPZ) and Dynasore also impaired IAV infection (S2C Fig), strongly supporting that IAVs undergo endocytosis via the CME pathway. Additionally, pre-culturing of A549 cells with Ferric ammonium citrate (FAC), iron chelating agent Deferoxamine (DFO), or TfR1 degradation compound Ferristatin-II indicated that FAC pre-culture reduced TfR1 expression and subsequently lowered susceptibility to IAV infection, while DFO pre-treatment increased TfR1 expression and enhanced susceptibility to IAV infection (S3A Fig). Notably, Ferristatin-II, known for its ability to degrade TfR1 [35], exhibited a significant antiviral effect in a mouse model (S3B Fig). We thus concluded that iron-dependent TfR1 levels correlate with viral entry through the TfR1-TTP-CLTA axis.

## 2.3. Identify TfR1-interacting viral components and their binding manner and sites

Subsequently, co-immunoprecipitation experiments were conducted using lysates from infected A549 cells to identify viral components interacting with TfR1. We found viral NA and M1 proteins were selectively co-immunoprecipitated with the anti-TfR1 antibody at either 2, 6 or even 24 hours post-infection, whereas HA and M2 proteins didn't exhibit such interaction (Figs 3A and S3F). These interactions between TfR1 and NA, as well as M1, were further confirmed through co-immunoprecipitation using anti-NA and anti-M1 antibodies. In contrast, neither anti-HA nor anti-M2 antibodies co-immunoprecipitated with TfR1. Furthermore, surface plasmon resonance (SPR) experiments reinforced these findings, indicating that TfR1 interacts with M1 and NA with KD values of 75.9 nM and 42.4 nM, respectively; no significant affinity was observed between TfR1 and HA or M2 (Fig 3B). Additionally, the binding affinity between TfR1 and immobilized IAVs on an SPR chip disclosed a KD value of 34.6 nM, closely resembling the KD value observed for NA-TfR1 binding (Fig 3B), supporting that the primary interaction of IAV with TfR1 predominantly occurs through its envelope NA. These findings strongly support the notion that the primary interaction of IAV with TfR1 predominantly occurs through the viral envelope protein NA rather than HA.

Given that TfR1 is a homodimeric transmembrane glycoprotein with its extracellular domain divided into apical (A), helical (H), and protease-like (P) domains, we sought to determine which domains might interact with NA. Incubation of IAVs with Tf protein revealed that a five-fold excess of Tf had minimal impact on IAV entry (S3C Fig) and also maintains iron delivery (S3D Fig), despite Tf's high affinity for TfR1 (KD = 0.72 nM). This suggests that the binding site for Tf on TfR1, primarily located in H and P domains, differs from that for IAVs. In contrast, co-incubation of recombinant TfR1 extracellular domain (TfR1$^{ECD}$) significantly inhibited IAV entry (S3C Fig), emphasizing TfR1's potential as an IAV receptor. Furthermore, a TfR1-specific antibody (OKT-9, targeting amino acid sequences 511–751, S3D Fig) displayed a partial inhibitory effect on IAV entry (S3C Fig), suggesting a partial overlap between the antibody's binding site and the virus binding site.

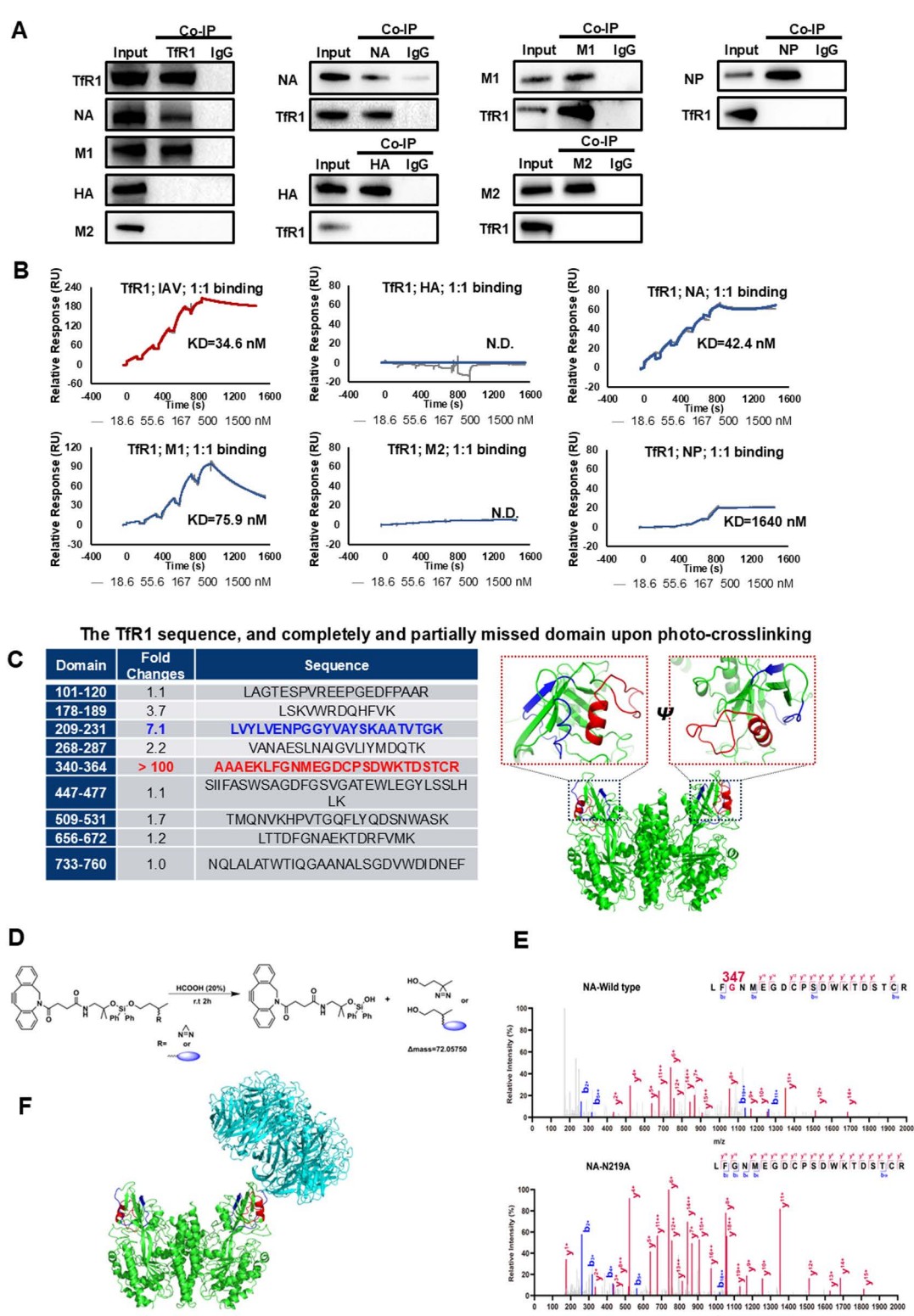

**Fig 3. Exploration of TfR1-interacting viral components, the binding manner and photo-crosslinked sites. (A)** Identification of viral components interacting with TfR1 through co-immunoprecipitation assay and western blotting. Lysates from A549 cells at 2 hours post-infection with IAV (MOI = 20) were co-immunoprecipitated using anti-TfR1 antibody along with anti-viral antibodies. **(B)** Characterization of TfR1's affinity with IAV particles and its

constituents using Surface Plasmon Resonance (SPR) experiments. IAV particles or viral proteins were immobilized on CM5 chips and interacted with TfR1 protein to measure the binding kinetics and affinity constants. **(C)** Identification of crosslinked sequences within TfR1 upon photo-activation using DIBO-DAZ-Biotin-labeled IAV particles. TfR1 protein was exposed to UV irradiation after incubation with the photo-crosslinking probe-labeled IAVs. UPLC-MS analysis was conducted to detect lost peptides (due to the covalent linkage of glycans and probe) within TfR1 marked in blue or red, with a control using IAV-N$_3$ (-) instead of DIBO-DAZ-Biotin labeled IAVs (+). **(D)** Schematic representation of a cleavable photo-crosslinking probe featuring an acid-susceptible diphenyl silane moiety between DIBO and Diazirine, facilitating the addition of a group with a precise molecular mass of 72.05750 in crosslinked TfR1 peptides upon photo-activation and folic acid treatment. **(E)** Identification of photo-crosslinked sequences and specific crosslinked sites within TfR1 via high-resolution mass spectrometry (HRMS), according to the precise molecular marker after acid-cleavage. TfR1 protein was exposed to UV irradiation after incubation with the cleavable photo-crosslinking probe-labeled IAV or its mutant version NA-N219A. UPLC-HRMS was utilized to identify the crosslinked peptides post-SDS-PAGE separation. **(F)** Structural representation of NA binding within the TfR1 protein (PDB: 6Q23 and 1SUV) derived from blind docking calculations. The final docked conformations of the NA tetramer (cyan) bound to the TfR1 dimer (green) were based on most suitable conformation and crosslinked sites. The proposed binding sites of NA occupy the apical domain of TfR1, considering a 3 Å resolution X-ray structure. TfR1's extracellular domain is categorized into apical domain **(A)**, helical (H) domain, and protease-like domain **(P)**, with primary transferrin binding in H and P domains. The NA tetramer binds to the apical domain of the TfR1 dimer, proximal to the crosslinked peptide of TfR1.

Regional deletion and site-specific mutation experiments revealed that deletions in the apical domain (Δ121–160/Δ181–220/Δ331–370) reduced susceptibility to IAV infection in TfR1-transfected CHO cells, while transfections of TfR1 variants (Δ481–520/Δ631–670/Δ721–758 in other domains) led to no decreased IAV uptake (S3E Fig), supporting the apical domain as the binding site for IAVs. Given TfR1 as one crosslinked protein upon radiation of IAV-DAZ-Biotin WSN particles with A549 cells, we then conducted a parallel photo-crosslinking experiment with TfR1 replacing A549 cells to identify which domains and even residues of TfR1 were engaged with WSN particles. Analysis of photo-crosslinking samples derived from the IAV & TfR1 complex, compared to the non-photo-crosslinking controls, using Q Exactive HF-X MS (Thermo) revealed a loss in peptides within the apical domain, spanning residues 209–231 and 340–364 (Fig 3C). This suggests that the apical domain of TfR1 may have been covalently crosslinked with WSN virus (NA). In order to identify photo-crosslinking sites, probe incorporating a cleavable diazirine moiety (Fig 3D) was utilized, which was designed to undergo acid-induced cleavage (10% formic acid), generating a mass tag (75.05750 Da) on the crosslinked proteins. High-resolution mass spectrometry (HRMS) revealed a precise 72.05750 Da increase in molecular weight, matching the crosslinked diazirine moiety, pinpointed to the peptide sequence 345-LFGNMEGDCPSDWKTDSTCR-364 (calc. $[M+H]^+$=2448.02695 Da, found $[M+H]^+$=2448.04502 Da, Δmass=0.01807 Da and Δppm=7.38, Fig 3E). Using this refined approach, we precisely mapped the TfR1-NA interaction to residues 345–364 within the apical domain (sequence: LFGN-MEGDCPSDWKTDSTCR). Critically, the diagnostic mass tag localized to G347 and N348 (calc. $[M+H]^+$=2448.02695 Da, found $[M+H]^+$=2448.04502 Da, Δmass=0.01807 Da and Δppm=7.38), confirming these residues as direct crosslinking sites. This component within TfR1's apical region features crosslinking sites specifically within or in proximity to residues G347 and D352 (S4A Fig). Concerning the potential glycosylation site of NA (N44, N72, N219) for the decorating probe, site-specific mutation and infectivity experiments showed that only the N219A variant displayed a discernible reduction in viral infection (S4B Fig), highlighting the crucial role of NA glycosylated N219 residue in its interaction with TfR1. Consistently, the crosslinked peptide spanning from 345 to 364 was detected in NA mutations N44A and N72A but absent in IAV featuring the N219A mutation (Figs 3E and S4A). ELISA results also confirmed this observation, demonstrating a noticeable decrease in binding affinity between the N219A variant and TfR1, which has been immobilized (S4B Fig). Furthermore the OKT-9 antibody attenuated IAV endocytosis by specifically binding to the apical domain of TfR1 (S4C Fig).A comparison of the KD value of TfR1 to WT NA vs. to N219A disclosed a substantial increase trend, supporting the binding of TfR1 to NA potentially via the N219 residue (S4D Fig). Building upon these insights, a docking analysis [38] of NA protein (PDB:6Q23) in complex with TfR1 protein (PDB:1SUV) vividly illustrated their binding at the TfR1 apical domain (Fig 3F and S1 Movie). Interactions included the formation of five hydrogen bonds between Lys86, Asn219, Glu243, Gln292 in NA and Ser370, Gly347, Asn348 in TfR1, respectively, and electrostatic interactions between Lys86, Asn219, Arg241, Lys244, Asn293, Glu295 in NA and Asp204, K371, Glu350, Gly347 in TfR1. Such a binding manner was supported by Julia Lederhofer's recent findings that NA-specific antibodies targeting this domain inhibit viral propagation of a wide

range of human influenza viruses [39]. Our findings elucidated the binding pattern between TfR1 and NA, explaining why antibodies specific to this dark side contribute to a broad-spectrum and effective antiviral strategy.

## 2.4. Explore TfR1-M1 interaction and implications for viral nucleocapsid uncoating

Given the robust affinity identified between TfR1 and matrix M1 in co-immunoprecipitation assay, a time-course analysis was employed to monitor the expression dynamics of M1 and TfR1 in infected A549 cells (MOI = 0.01, a lower MOI was used to allow the cells to survive under prolonged viral infection conditions). A gradual decrease in intracellular TfR1 levels coinciding with a steady increase in M1 expression was observed. Notably, TfR1 eventually became undetectable as M1 levels escalated, suggesting a depletion of TfR1 concurrent with an overwhelming abundance of M1 (Fig 4A). This effect was not observed in other membrane proteins such as EGFR and LDLR, nor at TfR1 mRNA levels (S5A Fig), ruling out cell stress as the likely cause but hinting at the involvement of a specialized protein degradation mechanism targeting TfR1. Subsequently, in the experiments where we either fixed the amount of TfR1 plasmid transfection and gradually increased the co-transfected M1/PB2 plasmid amount, or fixed the M1/PB2 plasmid transfection amount and gradually increased the co-transfected TfR1 plasmid amount, we demonstrated that elevating TfR1 expression in host cells led to a dose-dependent decrease in M1, but not PB2, protein levels (Figs 4B and S5B). More importantly, upon transfection with individual plasmids from the IAV plasmid complex, only the one encoding M1 protein demonstrated the capability to downregulate TfR1 expression (Fig 4C). This supports the notion of a mutually exclusive relationship between TfR1 and M1 protein, consistent with their direct binding.

Building upon prior research indicating the involvement of the host proteasome and aggresome systems in M1 degradation [31], we tested whether these pathways influence the regulation of TfR1 levels. Transfection of the IAV plasmid complex and TfR1 into 293T cells in the presence of MG132, a proteasome inhibitor, indicated a significant rescue in the expression levels of both TfR1 and M1, while HA and NA proteins remained unaffected (Fig 4D). Additionally, Tubacin, an HDAC6 inhibitor, but not the lysosome inhibitor HCQ, also elicited a notable increase in TfR1 and M1 levels, although to a lesser extent than the effect seen with MG132 (Figs 4E, S5C and S5E). To further validate these findings, parallel experiments in IAV-infected cells treated with Tubacin revealed viral titer patterns by qRT-PCR that were consistent with those observed in IAV plasmid-transfected HEK293T cells (Figs 4E and S5D). The significant inhibition of MG132 followed by Tubacin but not HCQ suggests engagement of the proteasome pathway and HDAC6 but not lysosome in this degradation process. Subsequent laser confocal microscopy experiments unveiled that co-transfected TfR1 and M1 formed protein aggregates within host cells (Fig 4F–G), co-localizing with ubiquitin and HDAC6; this was supported by the in vitro aggregation study (Fig 4H). From these observations, we infer that the mutual degradation of interacting M1 and TfR1 proteins involves an interplay between HDAC6-mediated protein degradation and ubiquitin-proteasome pathways, and the formation of the M1-TfR1 as a misfolded complex triggers their concurrent degradation.

## 2.5. Pinpoint TfR1 region responsible for triggering matrix protein degradation

To identify whether the extracellular or intracellular domain of TfR1(Fig 5A) is responsible for M1 degradation, we utilized SLC3A2, a membrane receptor that does not induce M1 degradation (S5F Fig). Meanwhile co-transfection of the M1 plasmid with TfR1 or SLC3A2 plasmid had no influence on M1 transcriptional level (S5G Fig). We created SITE by fusing the extracellular domain of TFR1 with the intracellular domain of SLC3A2 and TISE by fusing the intracellular domain of TFR1 with the extracellular domain of SLC3A2. Co-transfection of the M1 plasmid with TISE still resulted in M1 degradation, while repeating the co-transfection with SITE led to no M1 degradation (Figs 5B and S5H), indicating the intracellular domain of TfR1 as the trigger for M1 degradation. A FLAG-tag has been engineered into the chimeric constructs as an expression control. Consistently, TfR1 truncations (S5I Fig) showed that truncating any fragment within the extracellular domain had no impact on M1 degradation (S6A Fig), while removal of the transmembrane domain, either in conjunction

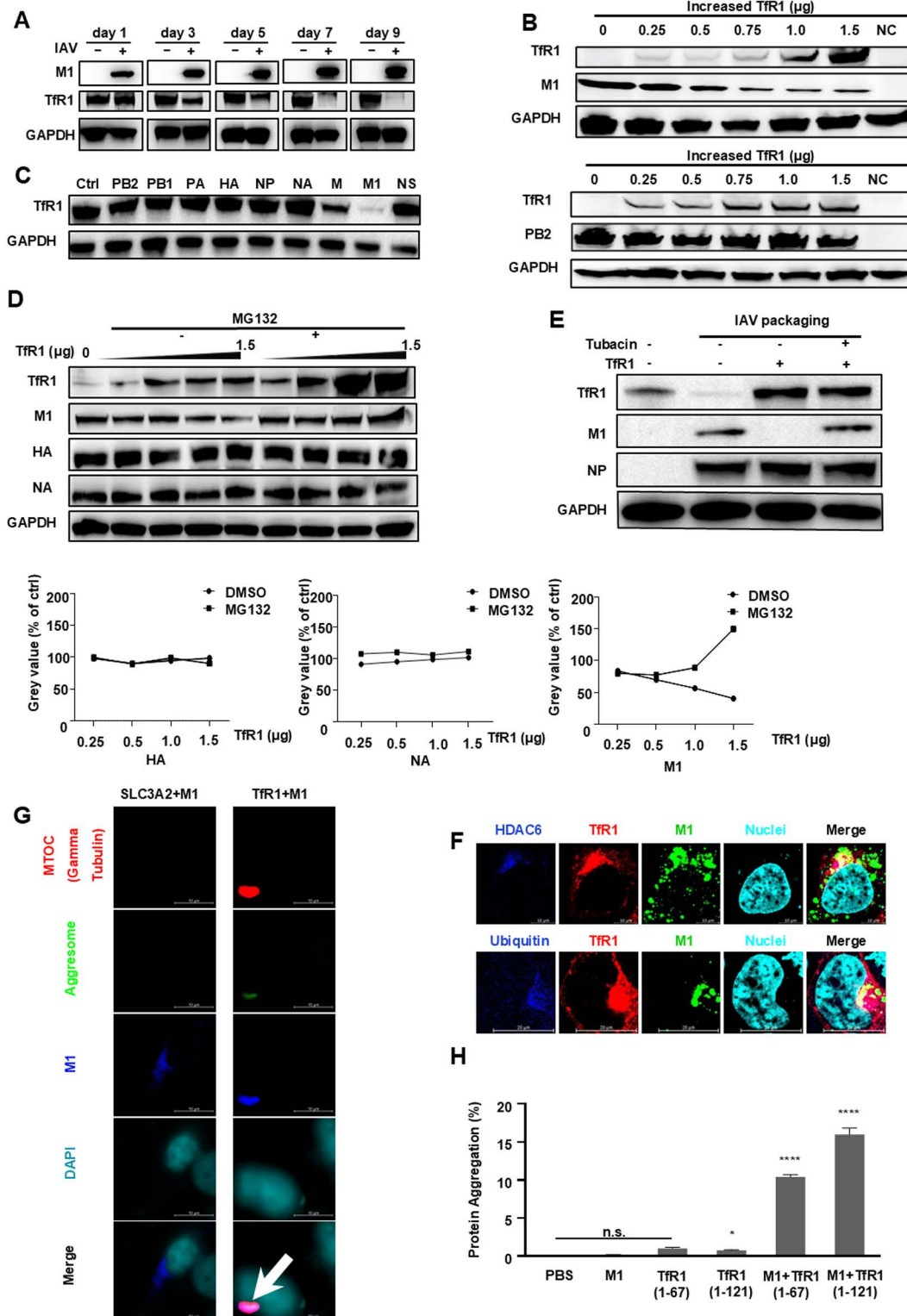

**Fig 4. Exploration of TfR1-M1 interaction and implication for viral uncoating. (A)** Time-course monitoring of viral M1 and host TfR1 expression levels in infected A549 cells (MOI = 0.01) conducted via western blotting analysis. The culture medium was refreshed daily, and cells were harvested at specified time points. Equal loading was confirmed by detecting GAPDH. **(B)** Assessment of the impact of TfR1 on viral M1 protein by co-transfecting the

M1 or PB2 plasmid (0.5 μg) with varying concentrations of TfR1 plasmid in 293T cells for 48 hours. Equal loading was confirmed by detecting GAPDH. **(C)** Determination of the specificity of TfR1-mediated viral protein degradation by individually co-transfecting plasmids from the IAV plasmid complex with the TfR1 plasmid in A549 cells for 48 hours. Equal loading was confirmed by detecting GAPDH. **(D)** Investigation into the role of the proteasome system in TfR1-mediated degradation of M1 protein. 293T cells were transfected with the IAV plasmid complex along with increased concentrations of TfR1 plasmid in the absence or presence of MG132 (10 μM). Analysis of TfR1 and different IAV proteins' protein levels was conducted via western blotting 48 hours post-transfection. Equal loading was confirmed by detecting GAPDH. Mean gray value were used to statistical analyze. **(E)** Exploration of the aggresome systems' role in TfR1-mediated degradation of M1 protein. 293T cells were co-transfected with TfR1 plasmid and the viral plasmid complex containing M1 and NP genes in the absence or presence of Tubacin (5 μM). Western blotting was performed to analyze TfR1, M1, and NP protein levels 48 hours post-transfection. Equal loading was confirmed by detecting GAPDH. **(F)** Laser confocal microscopy experiments to analyze cellular colocalization patterns involving HDAC6 (blue, upper) and ubiquitin (blue, lower) with TfR1 (red) and M1 (green) in TfR1/M1-ZsGreen transfected 293T cells. Scale bars, 10 μm. **(G)** Detection of aggresome (green)/MTOC(Gamma Tubulin, red) in 293T cells co-transfected with TfR1/SLC3A2 and M1 plasmids by laser confocal microscopy experiments using the Aggresome Detection Kit (ab139486). MTOC(Gamma Tubulin, red), M1 (blue), aggresome (green),nucleus(DAPI,cyan). The arrow indicates aggresome. Scale bars, 10 μm. **(H)** In vitro quantification of truncated TfR1, M1 and their complex protein aggregation by Protein Aggregation Assay Kit (ab234048). Aggregation was determined based on the ΔRFU using a standard curve for calculation. Unpaired t-tests were used for statistical analysis, denoting significance as follows: *p<0.05; **p<0.01; ***p<0.001; and ****p<0.0001, while "n.s." indicates non-significance.

with the extracellular or intracellular domain, abolished M1 degradation (Figs 5C and S6E), indicating the essential role of the transmembrane domain in TfR1-mediated M1 degradation. Additional truncation analysis revealed that the intracellular and transmembrane domains (Intra+Tm (1–88)) retained M1 degradation activity (Figs 5C and S6A). Successive removal of the N-terminal residues still led to a significant M1 reduction and this effect persisted until deletion of residue 53; further deletions beyond residue 54, i.e., the transfer-stop sequence adjacent to the transmembrane domain, diminished its capability to degrade M1 and removal of residues 1–63 entirely eliminated its degradation capability (Fig 5D). In a parallel experiment evaluating the effect of TfR1 truncation on viral entry, we found that none of the truncated variants restored the susceptibility of CHO cells to IAV infection (S6D Fig). Clearly, it is the intracellular transfer-stop sequence together with the transmembrane domain of TfR1 that triggers viral matrix M1 degradation. A docking analysis of M1 protein in complex with TfR1 protein supports the potential binding of IAV's M1 protein to the intracellular stop-transfer sequence of TfR1, as depicted in Fig 5E.

Given the IAV matrix protein as an important component for progeny virus production, TfR1-mediated degradation of M1 provides an exciting avenue for the development of novel antivirals. We first evaluated the broad-spectrum effect of TfR1-mediated degradation on the viral matrix protein, spanning antigenically distinct influenza strains. We found significant degradation of matrix proteins by TfR1 extended to A/Puerto Rico/8/1934 (H1N1), A/Aichi/2/1968 (H3N2), and A/Anhui/1/2013 (H7N9), attributed to 92% sequence homology in M1 among these distantly related influenza strains (Fig 6C). The influence of TfR1-mediated M1 degradation also displayed a clear dose-dependent inhibition of IAV progeny virus production from IAV plasmids complex-transfected cells and infected cells with IAV (Fig 6D). Furthermore, investigations revealed that TfR1-mediated degradation extended beyond influenza viruses, as co-transfection with TfR1 led to degradation of matrix proteins of measles virus (MeV) and rabies virus (RABV) in a dose-dependent manner (Figs S6C and 6E). We also tried to shorten TfR1 by substituting its extracellular domain and transmembrane sequence with an isoleucine zipper, and the intracellular domain was truncated into various versions (1–90-ZH, 1–63-ZH, and 53–67-ZH). It was found that all of them predominantly existed as dimers and trimers (Fig 6A) and exhibited notable efficacy in degrading M1 protein in co-transfection and IAV infection (Figs 6B and S6B), whereas the GST negative control showed no detectable impact on M1 degradation. Immunofluorescence analysis further confirmed the cytoplasmic localization of these truncation mutants similar to the full-length TfR1 (S6B Fig). This suggests that the role of the transmembrane domain is to maintain TfR1 as a dimer to trigger M1 degradation and can be replaced by an isoleucine zipper. Hence, the discovery of viral matrix protein degradation by TfR1 holds promise for the development of degradative antiviral agents that transcend the scope of influenza viruses.

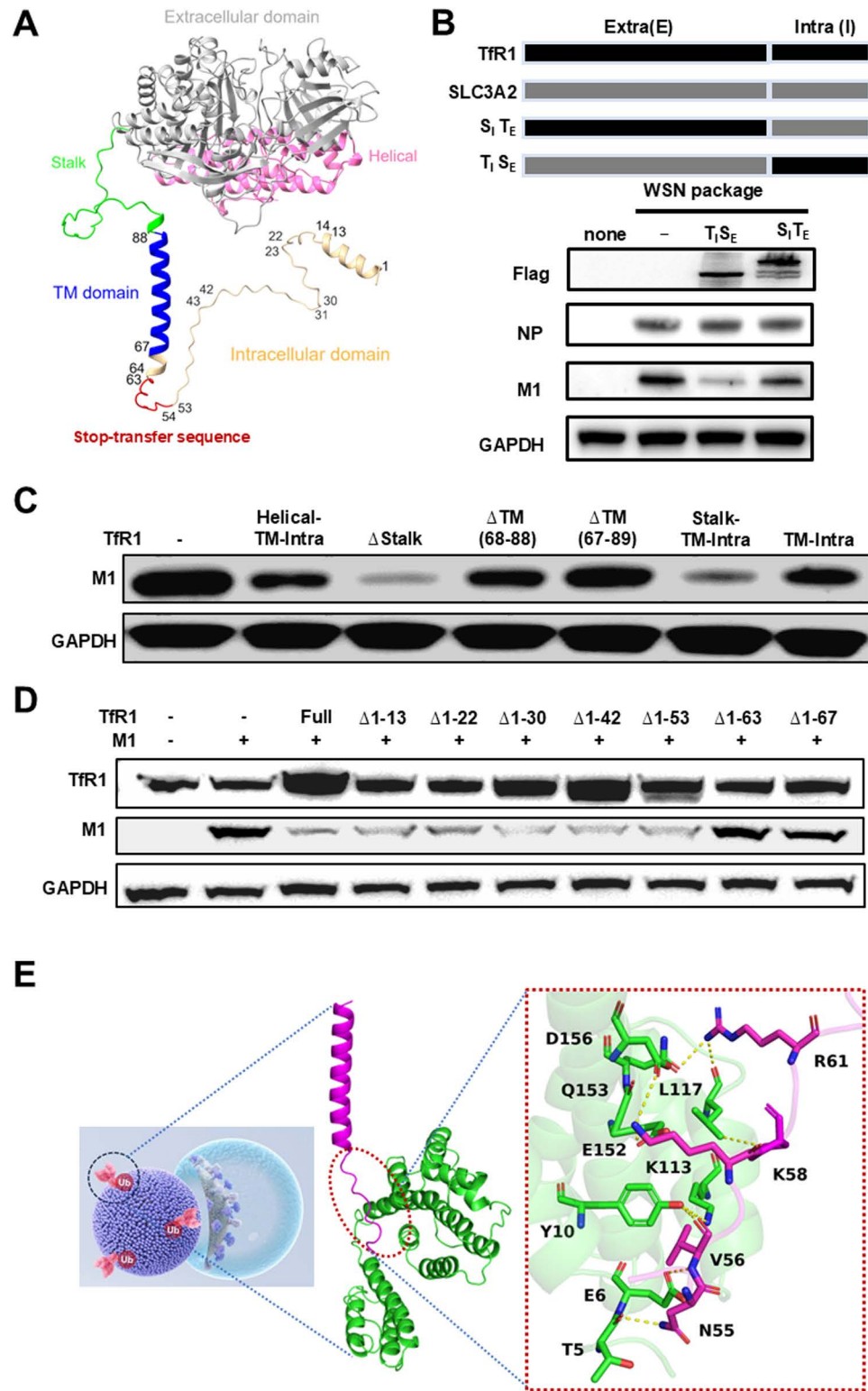

**Fig 5. Identification of TfR1 region responsible for triggering matrix protein degradation. (A)** Visualization of the predicted tertiary structure of TfR1 monomer generated by Alpha Fold, highlighting the transmembrane, stalk, and helical domains in blue, green, and purple, respectively. **(B)** Determination of the critical region in TfR1 responsible for initiating M1 degradation through domain swapping with SLC3A2. Co-transfection of resultant

TISE and SITE plasmids with the M1 plasmid in 293T enabled the reevaluation of M1 degradation, validated by western blotting. **(C)** Investigation into the contribution of TfR1's transmembrane domain in instigating M1 degradation. Truncated plasmids containing various sequences surrounding the transmembrane domain of TfR1, along with the M1 plasmid, were co-transfected into 293T cells for 48 hours. **(D)** Assessment of M1 degradation facilitated by various intracellularly truncated versions of TfR1. 293T cells underwent co-transfection with M1 plasmid and truncated TfR1 plasmids, followed by analysis of TfR1 and M1 protein levels through western blotting 48 hours post-transfection. **(E)** Structurally depicting the binding of M1 (green, PDB: 6Z5L) with the transmembrane and intracellular stop-transfer sequence of TfR1 (purple, AlfaFold2-prediction) based on blind docking calculations. TfR1 (residues 53-67) was proposed to occupy the NTD of M1, forming four hydrogen bonds between Thr5, Leu117, Gln153, Asp156 of M1 and Asn55, Arg61, Lys58 of TfR1, respectively, and a charge interaction between M1's Glu152 and TfR1's Lys58.

## 3. Discussion

The utilization of TfR1, the key regulator of iron metabolism [40], as a cellular entry point by many viruses, such as HCV [34], MACV [41], RABV [42] and so on [33,43,44], has been inadequately explored. The recent discoveries of the involvement of TfR1 in the entry of IAVs [45] and SARS-CoV-2 [46] add these life-threatening RNA viruses to this emerging enigma. Conventional methodologies for discovery and validation of viral receptors as well as viral-host interacting proteins include time- and cost-intensive genome-wide RNAi and CRISPR screening [47,48]. The pioneering work by Schmolke team on exploring TfR1 as a potential receptor of IAV entry reported a novel strategy that utilizes biotin proximity ligation on the envelope protein HA in its trimeric format [24], finding that the binding of IAV virions to TfR1 via sialic acids triggers IAV entry.

Here, we present a novel approach to decorate the sialylated envelope of IAVs with photo-crosslinking moieties, mimicking the metabolic introduction of terminal sialic acid from $Ac_4ManNAz$ in mammalian cells as previously reported [49]. This disclose a one stone two birds function of TfR1 exploited by influenza virus for IAV entry (Fig 6F). The azido group appended to the carbohydrate chain facilitates downstream bioorthogonal reactions [50], enabling the site-specific introduction of regular and cleavable photo-crosslinking moieties. Following photoactivation and crosslinking with host cells, a series of host factors implicated in IAV entry were identified, including the recently discovered TfR1 and previously identified EGFR, LDLR, and ITGA3 [20,51], validating our approach in exploring the multi-step process of RNA virus entry. The utilization of metabolic glycan engineering of IAVs allows for the simultaneous modification of HA and NA envelope proteins as well as other viral glycoproteins, providing a comprehensive tool to explore and capture intricate virus-host interactions. This contrasts with the approach utilizing biotin proximity ligation on HA [45], which overlooked the potential contributions of viral components other than HA in IAV entry. Thus, our approach complements the recently reported method.

It's widely recognized that HA plays a pivotal role in recognizing and engaging host glycoproteins, predominantly via the sialic acid moiety, during the initial phases of IAV entry [27,52–54]. The binding of HA to sialic acids on the epidermal growth factor receptor (EGFR), a member of the receptor-tyrosine kinase family induces EGFR clustering within lipid rafts so that EGFR activation triggers and facilitates IAV endocytosis in a sialic acid-dependent manner [55]. Recent study has disclosed that the binding of HA to TfR1 via sialic acids triggers IAV entry and TfR1 recycling is exploited by IAVs as a revolving door mechanism to enter host cells [45]. However, this finding appears to contradict to previous observation that sialic acid is not a requisite for effective IAV infection in GnT1-deficient CHO 15B and HEK 293S cells [25]. This aligns with previous studies that sialic acid only facilitates the initial viral attachment to the cell membrane but does not always guarantee efficient entry [26,27], implying a redundancy in entry mechanisms and the existence of other, yet unidentified, entry factors. Our observations suggest that the absence of TfR1 doesn't notably impact virus attachment but significantly affects uptake. Moreover, the impediment in viral protein import was particularly evident in cells lacking TTP or clathrin, mirroring the effect observed with TfR1 knockdown. Given TfR1 as an endocytosis membrane receptor for iron metabolism and, more importantly, the co-immunoprecipitation of TfR1 with NA rather than HA observed in this study, we hypothesize that TfR1 might serve as an anchoring point physically associated with incoming viruses, possibly through

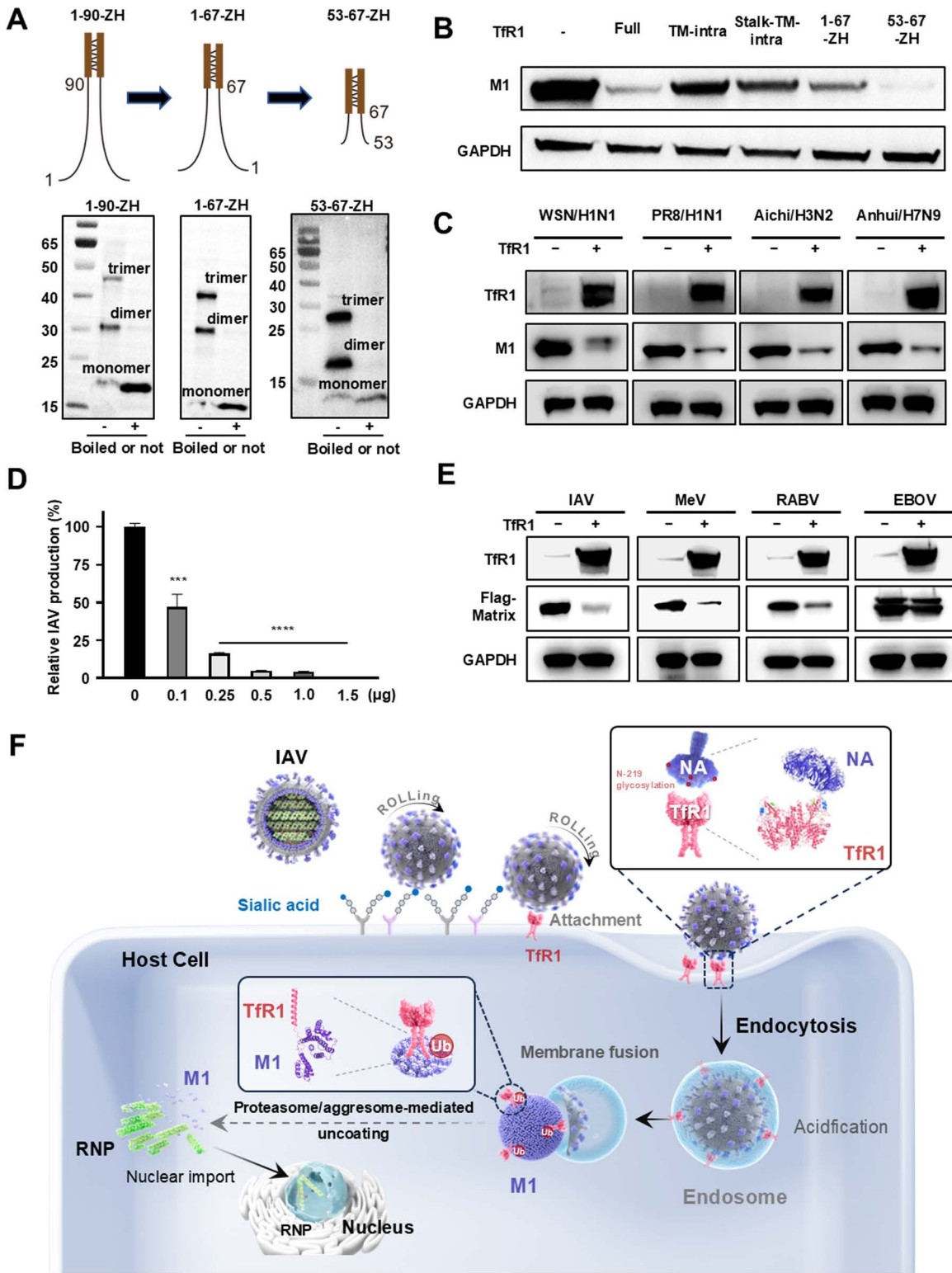

**Fig 6. Translational potentials of the discovery of mutual degradation of host receptor and viral matrix protein. (A)** Illustration depicting three truncated intracellular versions of TfR1 with an isoleucine zipper-his (ZH) instead of transmembrane domain, exhibiting predominantly homo-dimer and trimer formations, visualized through western blotting using anti-His antibody. **(B)** Investigation into the impact of the TfR1 transmembrane domain on

triggering M1 degradation. 293T cells underwent co-transfection with plasmids expressing M1 and truncated intracellular TfR1 variants featuring an isoleucine zipper-his instead of the transmembrane domain. Protein levels of M1 were assessed via western blotting 48 hours post-transfection. Equal loading was confirmed by detecting GAPDH. **(C)** Demonstration of the broad-spectrum effect of TfR1-mediated degradation on the viral matrix protein, spanning antigenically distinct influenza strains. 293T cells were co-transfected with TfR1 and M1 protein plasmids derived from different IAV strains. The protein levels of TfR1 and distinct M1 variants were analyzed via western blotting 48 hours post-transfection. Equal loading was confirmed by detecting GAPDH. **(D)** TfR1-mediated M1 degradation suppresses IAV progeny production in a dose-dependent manner. A549 cells were co-transfected with a total of 1.5 µg plasmid DNA, consisting of 12 IAV packaging plasmids and increasing amounts of TfR1 plasmid, balanced with the control plasmid SLC3A2 to maintain consistent total DNA across conditions. Viral titers were measured by RT-qPCR at 48 h post-transfection and normalized to untransfected controls. **(E)** Examination of the translational potential of TfR1-mediated degradation beyond influenza viruses. 293T cells were transfected with plasmids expressing TfR1 alongside matrix proteins from different viruses (MeV, RABV, and EBOV), each tagged with flag tag. Protein levels of TfR1 and various matrix proteins (Flag-tagged) were assessed via western blotting 48 hours post-transfection using anti-flag antibody. Equal loading was confirmed by detecting GAPDH. **(F)** Scheme representative of TfR1 -mediated endocytosis and uncoating. TfR1 functions as a receptor for kickstarting viral endocytosis by interacting its extracellular apical domain with the viral NA, bridging the gap between HA-sialic acid receptor attachment and clathrin-mediated viral endocytosis. Then, TfR1 acts as a trigger for matrix degradation by forming a misfolded protein complex with the viral matrix M1 via its intracellular stop-transfer sequence, filling the void between HA/endosomal fusion and proteasome/aggresome -mediated nucleocapsid uncoating. Unpaired t-tests were used for statistical analysis, denoting significance as follows: $*p < 0.05$; $**p < 0.01$; $***p < 0.001$; and $****p < 0.0001$, while "n.s." indicates non-significance.

interactions with viral NA, and subsequently initiates endocytosis upon virus attachment. It's likely that the combined interaction of NA with TfR1, following specific HA binding with sialic acid residues (Fig 6F), plays a pivotal role in defining IAV tropism and host specificity, given the low expression level of TfR1 in most tissues [56]. This may account for why only specific tissues and organs exhibit vulnerability to IAV infection.

Unlike HA, NA has long been associated primarily with virus budding, where it facilitates the removal of sialic acid bound to nascent viruses, enabling the release of viral particles [16,57]. However, our findings propose an additional role for NA in the early stages of IAV entry. Upon initial attachment of IAV to the cell surface via sialic acid groups, the virus navigates among sialic acid until encountering TfR1. In this process, NA plays a significant role by cleaving HA-bound sialic acid and, more importantly, by binding TfR1 and activating default clathrin-mediated endocytosis pathway (the TfR1-TTP-CLTA axis), similar to the case of TfR1 internalization induced by Tf binding (Fig 6F). The critical role of NA in guiding IAVs entry was supported by recent discovery that NA inhibitors can impede IAV entry [58,59], and antibodies against NA exert cross-protective effects within a subtype [60,61]. The slower drift of NA sequences than HA among seasonal influenza viruses makes NA an appealing target for antiviral therapies and vaccines. Although the crystal structures detailing NA-TfR1 binding remain elusive, our photo-crosslink data support the accessibility of residue N219, one of three N-glycosylation sites in NA (A/WSN/33 (H1N1) strain), to TfR1 within the apical domain, likely crossing at residues G347 or/and D352. This highlights the apical domain of TfR1 as an attractive target for exploration by IAVs and other viruses [41,62,63], as it permits viral binding without interfering with the protease-like and helical domains. Consistent with this structural information, the presence of Tf doesn't impede IAV infection and maintains iron delivery, essential for both the host and the pathogen.

Furthermore, the shared pathway for iron delivery utilized by several viruses offers convenience and potential advantages, enabling internalization into intracellular acidic compartments, pH-dependent cellular entry across endosomal domains, and access into the cytoplasm. After entry through the endocytic pathway, endosome pH decrease triggers HA's conformational change for fusion while also destabilizing viral matrix interactions [64–67], notably the histidine cluster formed by three sequential M1 monomers, for capsid disassembly [68–70]. Additionally, the degradation of M1 appears to be critical since M1 serves as a cytoplasmic retention signal, otherwise the M1-vRNP complex couldn't enter nucleus for replication [71]. Paradoxically, the rapid pH changes following the nucleocapsid escape through the fusion pore are likely to reverse the disassembly process unless aided by other factors [72]. Considering that the viral M1 specifically co-precipitated with TfR1 in this study, it's conceivable that TfR1 directly interacts with the nucleocapsid, which generates physical forces aiding in dissociating the M1 proteins and capsid disassembly. It is well-known that misfolded proteins in

eukaryotic cells can be modified with Ubiquitin (Ub) for destruction by proteasomes [73–75]; when the proteasome system becomes overwhelmed it can also activate the aggresome pathway via HDAC6 for degradation [76,77]. In this context, TfR1 binding M1 via its intracellular domain could result in M1 misfolding within infected cells, thereby activating the proteasome and aggresome pathways to alleviate misfolded protein-induced stress. This, in turn, might facilitate capsid uncoating through the mutual degradation of the host receptor and the viral matrix protein. Furthermore, degradation of M1 in infected cells, mediated by the formation of TfR1-M1 aggregates activating proteasome and aggresome pathways, promotes the nucleocapsid uncoating. This may explain why headless TfR1 was previously shown to still enhance virion entry in Schmolke's team work [24].

Notably, despite TfR1's capability to interact with NA and M1 through its extracellular and intracellular domains, respectively, only the M1 protein undergoes degradation, emphasizing the crucial role of TfR1's intracellular domain (possibly the stop-transfer sequence as elucidated by the deletion assay) in initiating viral matrix degradation. The TfR1-mediated degradation of M1 appears to be critical for viral RNA replication, since M1 serves as a cytoplasmic retention signal for the M1-vRNP complex while viral RNA replication occurs exclusively within the nucleus. This underscores the essential role of TfR1 as a matrix degradator, facilitating the release of viral RNAs into the nucleus. The revelation of mutual degradation between the host receptor and the viral matrix protein presents broader implications for the development of antiviral strategies, extending beyond IAVs. In this study, the TfR1-mediated degradation of M1 presents a potential advantage of being less prone to developing viral resistance, a challenge commonly encountered with M2 ion channel inhibitors. Given that M1 is a crucial structural element of capsids housing vRNPs for the generation of progeny viruses, it's intriguing to note that leveraging the intracellular domain of TfR1, even as a short peptide in dimer form, serves as a broad-spectrum viral matrix M1 degradator. Considering that other viruses might employ a similar mechanism or utilize TfR1 in degradation of their matrix proteins, this novel antiviral strategy could have applicability beyond IAVs. As observed in this study, such an approach may hold promise for combating other enveloped viruses, such as measles virus (MeV) and rabies virus (RABV).

There are also several limitations. While we have identified the direct interaction between TfR1 and IAV's NA/M1 proteins, the precise region of this interaction remains to be elucidated. The exact binding configuration, as informed by detailed structural analysis, has yet to be determined and warrants further rigorous investigation. Regarding the reciprocal degradation of TfR1 and M1, multiplexed mechanisms and factors rather than just proteasome- and aggresome-mediated pathway reported in this study may engage in the process. Detailed investigation into the molecular mechanisms is necessary and may have broad implications.

In summary, our approach, utilizing the decoration of the sialylated envelope of IAVs with photo-crosslinking moieties, sheds light on the significant role of TfR1 in virus entry. Initially serving as an endocytosis trigger by engaging its apical domain with the viral envelope NA, TfR1 subsequently functions as a matrix degradator, interacting via its intracellular stop-transfer sequence with the viral matrix M1. This dual functionality of TfR1 offers a plausible explanation for why and how a central regulator of iron metabolism is exploited by numerous viruses as a cellular entry point (Fig 6F). These insights present an exciting avenue for the development of innovative antiviral strategies even beyond IAVs. Furthermore, in addition to receptor-dependent viral entry, viruses may utilize alternative infection pathways, such as the dynamin-independent macropinocytic pathway [78–80], under different conditions. This highlights the complexity of viral infection mechanisms and underscores the need for further investigation.

## 4. Materials and methods

Cells: Human embryonic kidney (HEK) 293T cells, Madin-Darby canine kidney (MDCK) cells, human lung adenocarcinoma A549 cells, Human astroblastoma U87 cells and Chinese hamster ovarian CHO cells and Lec1 cells were obtained from American Type Culture Collection (ATCC). HEK293T, MDCK, A549 and U87 cells were cultured in Dulbecco's modified Eagle's medium (DMEM; Macgene, Beijing, China) with 10% fetal bovine serum (FBS; PAN Biotech,

Bavaria, Germany), 100 IU/mL penicillin, 1 µg/mL streptomycin. CHO and Lec1 cells were cultured in Dulbecco's Modified Eagle Medium/Nutrient Mixture F-12 (DMEM/F12; Beit-Haemek, Israel) with 10% FBS. All cells were cultured at 37°C with 5% $CO_2$.

Viruses: Influenza virus A/PR/8/1934 (H1N1) and Influenza virus A/Aichi/2/68 (H3N2) was kindly provided by Sinovac Biotech Ltd (Beijing). The 12-plasmid influenza virus A/WSN/33(H1N1) rescue system was donated by Professor Gao Fu, Institute of Microbiology, Chinese Academy of Sciences, Beijing, China.

Mice strains: Mice strains used in this study were Balb/c mice.

Antibodies and Proteins: Mouse anti-CD71 monoclonal antibody (clone 3C11F11, Proteintech, #66180–1-Ig), Rabbit anti-SH3 BP4 polyclonal antibody (Proteintech, #17691–1-AP), Mouse anti-GAPDH monoclonal antibody (1E6D9, Proteintech, #60004–1-Ig), Mouse anti-NP (Influenza A Virus) monoclonal antibody (9G8, Santa Cruz, #sc-101352), Mouse anti-HA (H1N1) monoclonal antibody (2D10C9E2, Sino Biological, #11052-MM06), Mouse anti-NA (H1N1) monoclonal antibody (GT288, GeneTex, t#GTX629696), Rabbit anti-matrix protein (M1, H1N1) polyclonal antibody (GeneTex, #GTX125928), Rabbit anti-matrix protein (M2, H1N1) polyclonal antibody (GeneTex, Inc (GeneTex, #GTX125951), HDAC6 Rabbit mAb (D2E5, Cell Signaling Technology, #7558), Ubiquitin Rabbit mAb (E4I2J,Cell Signaling Technology, #43124), DYKDDDDK tag mouse Monoclonal antibody (Proteintech, #66008–4-Ig), His-Tag Mouse Monoclonal antibody (Proteintech, #66005–1-Ig), Mouse FITC anti-human CD71 monoclonal Antibody (CY1G4, BioLegend, #334104), CD71 Mouse Monoclonal Antibody (OKT-9, Thermo Fisher, #14-0719-82), Anti-rabbit IgG, HRP-linked Antibody (Cell Signaling Technology, #7074), Rabbit anti-Biotin polyclonal antibody (Bioss, #bs-0311R), Anti-mouse IgG HRP-linked Antibody (Cell Signaling Technology, #7076), Goat Anti-Rabbit IgG H&L Alexa Fluor 488 (Abcam, #ab150077), Goat Anti-Rabbit IgG H&L Alexa Fluor 555 (Abcam, #ab150078), Goat Anti-Rabbit IgG H&L Alexa Fluor 594 (Abcam, #ab150080), Goat Anti-Mouse IgG H&L Alexa Fluor 488 (Abcam #ab150113), Goat Anti-Mouse IgG H&L Alexa Fluor 647 (Abcam, #ab150115), Alexa Fluor 488 conjugated-Streptavidin (Yeasen Biotechnology, #35103ES60), Streptavidin Agarose Resin 6FF (Yeasen Biotechnology, #20512ES60). Transferrin Receptor/TFR1/CD71 Protein, Human, Recombinant (ECD, His Tag), HPLC-verified (Sino Biological, Cat#11020-H07H), Influenza A H1N1 (A/WSN/1933) Hemagglutinin/ HA Protein (His Tag) (Sino Biological, Cat#11692-V08H), Influenza A H1N1 (A/Puerto Rico/8/34/Mount Sinai) Matrix protein 1/ M1 Protein (His Tag), (Sino Biological, Cat#40010-V07E). The full length TfR1 were customized purchased from Sino Biological.

Reagents: Transferrin (Harveybio, #IPN1066), Sulfo-NHS-SS-Biotin (Thermo Fisher, #21331), TCEP (Yeasen Biotechnology, #20330ES03), 4% paraformaldehyde (Solarbio, #P1110), RIPA lysis buffer (Solarbio, # R0020). Ferristatin II, Ferric ammonium citrate and Deferoxamine were purchased from Sigma-Aldrich. Transferrin Receptor protein and Influenza HA/NA/M1/M2/NP Protein were purchased from Sino Biological. $NH_4Cl$, Dynasore, Puromycin, Chlorpromazine hydrochloride, MβCD, MG132, Tubacin, HCQ, ULK1–101 were purchased from Merck.

Plasmid construction: TfR1 and Viral matrix (PR8/H1N1, Aichi/H3N2, Anhui/H7N9) expression plasmid was purchased from Sino Biological, Inc (HG11020-CH, VG40010-UT, VG40215-UT, VG40107-UT). SLC3A2 expression plasmid was purchased from ZOMANBIO (ZK6793). Viral matrix (EBOV/RABV/MeV/ RSV) expression plasmid were synthesized from GENEWIZ (Beijing, China). TfR1 truncation plasmids and NA point mutation plasmids were constructed through Q5 Site-directed Mutagenesis Kit (E0552S, NEB).

Synthesis of Ac$_4$ManNAz and photo-crosslinking probe: The compound mannosamine (300 mg, 1.39 mmol) and TEA (351 mg, 3.48 mmol) was dissolved in MeOH (10 mL) and chloroacetic anhydride (475 mg, 2.78 mmol) was slowly added. The mixture was stirred at room temperature overnight. Then the solvent was evaporated and the intermediate was purified by flash column chromatography (200–300 µM mesh, Qingdao Haiyang Chemical Co., Ltd., Qingdao, China). After that the intermediate was dissolved in pyridine (2 mL) and cooled to 0°C to add Ac$_2$O (0.5 mL) slowly. The solution was transferred to room temperature and allowed to stir overnight. After completion (TLC) the reaction solvent was evaporated and extracted with EtOAc and 1M HCl. Then the crude product was purified by column chromatography (PE:EA from 4:1–1:1) to afford Ac$_4$ManNAz a white foam in 60% yield; Rf = 0.60 (petroleum ether:EtOAc = 1:1); $^1$H NMR (400

MHz, CDCl$_3$): δ 6.58 (d, 1H, $J$ = 9.4 Hz), 6.03 (d, 1H, $J$ = 1.5 Hz), 5.32 (d, 1H, $J$ = 4.2 Hz), 5.24-5.03 (m, 2H), 4.61 (ddd, 1H, $J_1$ = 1.8 Hz, $J_2$ = 4.2 Hz, $J_3$ = 9.3 Hz), 4.26-4.20 (m, 2H), 2.17 (s, 3H), 2.10 (s, 3H), 2.05 (s, 3H), 1.99 (s, 3H); $^{13}$C NMR (100 MHz, CDCl$_3$): δ 170.5, 170.0, 169.5, 168.1, 166.7, 91.3, 70.2, 68.8, 65.1, 61.8, 52.4, 49.3, 20.8, 20.7, 20.6, 20.6.

The diazirine was synthesized based on previous literature [81]. Levulinic acid (8.19 g, 70.5 mmol) was added to 250-mL round-bottomed flask, and 70 mL of 7M NH3-MeOH was added under nitrogen protection. The mixture was stirred in ice-water bath for 3 hours. Hydroxylamine-O-sulfonic acid (9.16 g, 81.1 mmol) was dissolved in MeOH and added drop-wise. The reaction mixture was stirred at room temperature overnight. The white precipitate was filtrated and filtrate was collected, evaporated and redissolved in MeOH. 15 mL TEA was added and then 17.9 g of iodine was used to oxidize diaziridine. The solution was extracted with EtOAc and 1M HCl, followed by washing with 10% sodium thiosulfate and saturated aqueous sodium chloride. The organic layer was evaporated to afford diazirine as a yellow-orange oily liquid in 32% yield.

D-biotin (1.5 g, 6.2 mmol) was activated with N-hydroxy succinimide (1.1 g, 9.6 mmol) to obtain NHS-biotin. Then NHS-biotin (341 mg, 1 mmol) and N-BOC-ethylenediamine (176 mg, 1.1 mmol) was dissolved in 10 mL MeOH and DIPEA (390 mg, 3 mmol). The reaction was stirred at room temperature for 6 hours and evaporated, followed by column chroma-tography (DCM:MeOH from 20:1–10:1). After that BOC was removed by TFA/MeOH to afford 58 as a white solid in 92% yield. $^1$H NMR (400 MHz, CD$_3$OD): δ 4.94 (dd, 1H, $J_1$ = 4.3 Hz, $J_2$ = 7.8 Hz), 4.31 (dd, 1H, $J_1$ = 4.4 Hz, $J_2$ = 7.9 Hz), 3.24 (m, 2H), 3.15 (m, 2H), 2.93 (dd, 1H, $J_1$ = 4.9 Hz, $J_2$ = 12.7 Hz), 2.72-2.68 (m, 2H), 2.21 (m, 2H), 1.77-1.55 (m, 4H), 1.50-1.45 (m, 2H), 1.43 (s, 9H); $^{13}$C NMR (100 MHz, CD$_3$OD): δ 174.9, 173.2, 164.7, 157.2, 78.7, 61.9, 60.2, 55.5, 48.4, 39.6, 39.5, 39.1, 35.4, 28.3, 28.0, 27.3, 25.4, 24.8.

Compound 58 (360 mg, 1.2 mmol) and N$_\alpha$-BOC-Nε-Fmoc-L-Lysine (611 mg, 1.3 mmol) was dissolved in 15 mL dry DMF. HATU (497 mg, 1.3 mmol) and DIPEA (613 mg, 4.7 mmol) was added subsequently. The mixture was stirred at room tem-perature overnight and then purified by column chromatography (DCM:MeOH from 20:1–6:1). Then Fmoc was removed by 20% piperidine/DMF to obtain 59 as a white solid in 81% yield. $^1$H NMR (400 MHz, CDCl$_3$): δ 7.63-7.47 (m, 4H), 7.27-7.15 (m, 4H), 4.35-4.08 (m, 4H), 3.56-3.49 (m, 1H), 3.31-3.10 (m, 4H), 3.01-3.29 (m, 4H), 2.05-1.92 (m, 2H), 1.59-1.44 (m, 5H), 1.39-1.04 (m, 18H); $^{13}$C NMR (100 MHz, CDCl$_3$): δ 178.6, 177.7, 160.9, 160.7, 147.887, 147.6, 145.2, 145.1, 132.5, 131.6, 131.0, 129.0, 128.9, 123.8, 83.1, 81.4, 70.9, 65.6, 64.1, 59.4, 59.2, 58.1, 51.0, 46.3, 44.1, 43.7, 43.1, 43.0, 39.5, 35.6, 33.1, 32.2, 31.9, 29.3, 26.7, 22.2, 20.9, 16.0.

Compound 59 (630 mg, 1.1 mmol), HATU (498 mg, 1.2 mmol) and TEA (472 mg, 4.7 mmol) was added to dry DMF. Then diazirine (168 mg, 1.3 mmol) was added. The reaction was allowed to stir for 6 hours at room temperature and purified with column chromatography (DCM:MeOH from 20:1–10:1). The BOC was removed by TFA/DCM to afford 62 as a light-yellow solid in 52% yield. $^1$H NMR (400 MHz, CD$_3$OD): δ 4.50 (dd, 1H, $J_1$ = 4.6 Hz, $J_2$ = 7.8 Hz), 4.32 (dd, 1H, $J_1$ = 4.4 Hz, $J_2$ = 7.8 Hz), 4.20 (m, 1H), 3.38-3.33 (m, 6H), 3.03 (t, 3H), 2.93 (dd, 1H, $J_1$ = 5.0 Hz, $J_2$ = 12.8 Hz), 2.70 (d, 1H, $J$ = 12.7 Hz), 2.23-2.16 (m, 4H), 1.83-1.70 (m, 2H), 1.69-1.55 (m, 6H), 1.51-1.39 (m, 15H), 1.02 (s, 3H); $^{13}$C NMR (100 MHz, CD$_3$OD): δ 175.0, 173.5, 173.3, 157.2, 128.0, 120.3, 78.5, 61.8, 60.3, 55.5, 53.8, 48.4, 46.5, 39.6, 38.9, 38.6, 35.4, 31.2, 29.9, 29.6, 29.2, 28.2, 28.0, 27.4, 25.4, 25.0, 22.9, 18.4.

Compound 62 (320 mg, 0.6 mmol) and DIPEA (155 mg, 1.2 mmol) was dissolved in dry DMF. DIBO-ester (221 mg, 0.6 mmol) was gradually added. The mixture was stirred for 3 hours at room temperature and purified with column chroma-tography (DCM:MeOH from 30:1–10:1) to afford 63 (DIBO-DAZ-Biotin probe) as a yellow solid in 61% yield. $^1$H NMR (400 MHz, CD$_3$OD): δ 7.57-7.19 (m, 8H), 7.05 (s, 1H), 4.46 (dd, 1H, $J_1$ = 3.3 Hz, $J_2$ = 12.2 Hz), 4.49-4.44 (m, 1H), 4.29-4.17 (m, 2H), 3.74-3.68 (m, 1H), 3.35-3.13 (m, 9H), 2.68 (d, 1H), 2.23-2.10 (m, 4H), 1.68-1.53 (m, 10H), 1.42 (d, 2H, $J$ = 6.5 Hz), 1.00 (s, 3H); $^{13}$C NMR (100 MHz, CD$_3$OD): δ 176.4, 174.9, 174.7, 166.1, 158.0, 153.7, 152.5, 131.1, 129.4, 129.3, 128.7, 128.3, 128.3, 127.2, 126.9, 124.9, 122.4, 113.8, 111.0, 77.8, 63.2, 61.6, 56.9, 55.8, 55.1, 49.8, 47.9, 43.8, 41.0, 40.0, 36.8, 32.5, 31.2, 31.0, 30.5, 29.6, 29.4, 26.7, 24.1, 19.8. ESI-HRMS: Calc for C$_{40}$H$_{50}$N$_8$O$_6$S [M + H]$^+$: 771.3647; found 771.3665.

A 3-neck round-bottomed flask containing 4-hydroxy-2-butanone (3.1 g, 34.5 mmol, 1.0 eq) was cooled to 0°C and 7 N NH$_3$ in MeOH (35 mL) was added slowly under N$_2$. After 3 hours, NH$_2$OSO$_3$H (4.25 g, 37.5 mmol, 1.1 eq) was dissolved in anhydrous methanolic and added dropwise to the reaction at 0°C. The reaction mixture was allowed to warm to room temperature overnight. The mixture was evaporated to dryness by blowing with N$_2$, and the remaining reside was dissolved in MeOH, and the insoluble material was filtered away. The filtrate was concentrated under reduced pressure and re-dissolved in anhydrous MeOH (25 mL). The solution was cooled to 0°C, and Et$_3$N (7.5 mL) was added. I$_2$ was then added in small portions until a dark brown colour persisted for more than 10 minutes. The solution was concentrated under reduced pressure and diluted with EtOAc. The organic phase was washed successively with 1 N HCl, sat. aq. Na$_2$S$_2$O$_3$ and then dried over anhydrous MgSO$_4$, the result solution was concentrated under reduced pressure to afford WMY-01–41-PP as a pale yellow oil. R$_f$ = 0.35 (petroleum ether: EtOAc = 1:1). $^1$H NMR (CDCl$_3$, 400 MHz): δ 3.54 (t, 2H, $J$ = 6.3 Hz), 1.64 (t, 2H, $J$ = 6.3 Hz), 1.45 (m, 1H), 1.08 (s, 3H).

To a solution of DBCO (200 mg, 0.64 mmol), HOBt (17.6 mg, 1.28 mmol, 2.0 eq) and EDCI (248 mg, 1.28 mmol, 2.0 eq) in DMF (5 mL), 1-amino-2-methyl-propan-2-ol (27 mg, 0.31 mmol) was added, followed by DIEA (38 µL, 1.92 mmol, 3.0 eq). After 4 hours, the solvent was evaporated under reduced pressure and the residue was taken up in CH$_2$Cl$_2$ and purified by silica gel chromatography with a 15–20% gradient of MeOH in CH$_2$Cl$_2$ to give compound WMY-01–110 as brown oil. R$_f$ = 0.60 (CH$_2$Cl$_2$: CH$_3$OH = 10:1). $^1$H NMR (CDCl$_3$, 400 MHz): δ 7.99 (s, 1H), 7.65 (d, 1H, $J$ = 7.4 Hz), 7.21-7.41 (m, 5H), 6.30 (s, 1H), 5.12 (d, 1H, $J$ = 13.9 Hz), 1.06 (s, 3H), 3.65 (d, 1H, $J$ = 13.8 Hz), 3.15 (dd, 1H, $J$ = 13.8, 6.0 Hz), 3.01 (dd, 1H, $J$ = 13.8, 6.0 Hz), 2.79 (ddd, 1H, $J$ = 16.9, 8.8, 5.2 Hz), 2.39 (ddd, 1H, $J$ = 14.1, 8.8, 5.1 Hz), 1.97 (dt, 1H, $J$ = 14.7, 6.0 Hz), 1.11 (s, 3H); $^{13}$C NMR (100 MHz, CDCl$_3$): δ 173.33, 172.41, 151.15, 148.00, 132.26, 129.30, 128.67, 128.26, 128.16, 127.74, 127.10, 125.53, 123.20, 122.49, 114.64, 107.81, 55.62, 50.36, 30.59, 27.15, 26.97; ESI-HRMS (m/z) Calcd for C$_{23}$H$_{25}$N$_2$O$_3$ [M + H]$^+$: 377.1865. Found 377.1860.

Triethylamine (125 µL, 0.90 mmol, 9.0 eq) and dichlorodiphenylsilane (63 µL, 0.30 mmol, 3.0 eq) were added in sequence to a stirred solution of WMY-01–110 (36 mg, 0.10 mmol, 1.0 eq) in dichloromethane (2 mL) and stirred for 2 h at room temperature. Then 2-(3-methyl-3H-diazirin-3-yl)ethan-1-ol (125 mg, 0.60 mmol, 6.0 eq) was added and stirred for an additional 1 hour at 24°C. The product mixture was diluted sequentially with dichloromethane (10 mL) and saturated aqueous sodium bicarbonate solution (10 mL). The resulting biphasic mixture was transferred to a separatory funnel and the layers that formed were separated. The aqueous layer was extracted with dichloromethane (3 × 10 mL), and the organic layers were combined and dried over sodium sulfate. The dried solution was filtered, and the filtrate was concentrated. The residue obtained was purified by flash-column chromatography (eluting with 1% methanol–dichloromethane, grading to 10% methanol-dichloromethane, 3 steps) to afford the diphenyl silane probe (15.4 mg, 43%) as pale-yellow oil. Rf = 0.50 (CH$_2$Cl$_2$: CH$_3$OH = 20:1). $^1$H NMR (400 MHz, CDCl$_3$): δ 7.60-7.64 (m, 5H), 7.46-7.50 (m, 1H), 7.21-7.45 (m, 12H), 6.00 (t, 1H, $J$ = 5.7 Hz), 5.13 (d, 1H, $J$ = 13.8 Hz), 3.61-3.70 (m, 3H), 3.21 (m, 2H), 2.79 (ddd, 1H, $J$ = 16.7, 8.3, 6.4 Hz), 2.41 (ddd, 1H, $J$ = 14.9, 8.2, 6.4 Hz), 2.07 (dt, 1H, $J$ = 15.1, 6.2 Hz), 1.93 (dt, 1H, $J$ = 16.8, 6.2 Hz), 1.54 (t, 2H, $J$ = 6.3, 6.0 Hz), 0.99 (s, 3H), 1.16 (s, 3H); $^{13}$C NMR (100 MHz, CDCl$_3$): δ 172.11. 171.98, 151.37, 148.04, 134.78, 133.82, 133.77, 132.26, 130.27, 129.30, 128.57, 128.12, 128.01, 127.86, 127.64, 126.99, 125.39, 123.16, 122.41, 114.68, 107.85, 75.74, 58.19, 55.44, 50.33, 37.10, 31.27, 30.16, 27.61, 27.33, 24.23, 20.10; ESI-HRMS (m/z) Calcd for C$_{39}$H$_{41}$N$_4$O$_4$ [M + H]$^+$: 657.2897. Found 657.2900.

Virus package and metabolic glycan labelling: HEK29T cells were seeded to 6-well plate at a density of 3 × 10$^5$ cells per well. Influenza A virus package was co-transfected with 12 plasmids (pHH21-PB2, pHH21-PB1, pHH21-PA, pHH21-HA, pHH21-NP, pHH21-NA, pHH21-M, pHH21-NS, pcDNA-PB2, pcDNA-PB1, pcDNA-PA, pcAGGS-NP) and transfection reagent PEI. 6 hours after transfection, medium was changed to DMEM containing 1% FBS and 0.2 µg/ mL TPCK-trypsin. After the cytopathic effect occurred in more than 90% of the cells, the supernatant of the cells was collected, centrifuged at 1000g for 5 min to remove cell debris and filtrated using 0.45 µm filter membrane. Freezing the virus at -80°C for subsequent experiments. For further purification, virus was ultracentrifuged at 100, 000 × g for 6 hours to remove supernatant and virus particles was resuspended in PBS.

For metabolic glycan labelling, 100 μM $Ac_4ManNAz$ was added when changing medium after transfection. $Ac_4Man$-NAz would be transformed into sialic acid derivative and transferred to the end of N-glycan during virus package in host cells. The medium was collected and concentrated by sub-ultracentrifugation to obtain purified virus particles. The azido-labelled influenza virus was reacted with DIBO-DAZ-Biotin probe (63, 100 μM) at 4°C overnight, followed by ultrafiltration to removed excess probe.

UV photo-crosslinking: A549 cells were seeded into $10\,cm^2$ dishes at a density of $3 \times 10^6$ cells 12 hours before photo-crosslinking. Then host cells were infected with DIBO-DAZ-Biotin labelled Influenza A virus (MOI = 50, to increase the capture rate of photo-crosslinking) and incubated at 4°C for 2 hours. The medium was removed and cells were washed with chilled PBS for three times, followed by 365nm-UV irradiation on ice for 20min. No-UV-irradiated sets and blank control sets were placed on ice in dark. All UV-irradiated sets, non-UV-irradiated sets and blank control sets cells were harvested and lysed with precooling RIPA lysis buffer respectively. Cell lysates were centrifuged (14, 000 × g, 20 min) to collect the supernatant protein. The photo-crosslinked protein was captured and enriched by incubation of 100 μL Streptavidin agarose for 6 hours at 4°C in dark. The agarose was washed with PBS for five times and then eluted by boiling 10min with SDS-PAGE loading buffer.

In the context of cleavable photo-crosslinking, TfR1 protein and the diphenyl silane probe, specifically labelled with IAV, were conjoined in PBS and incubated at 37°C for a duration of 30min. Following this, the samples underwent irradiation employing a 365nm UV light source or were maintained in darkness on ice for a comparable 30min period. Post-incubation, the samples were treated with formic acid, achieving a final concentration of 20%, at ambient temperature for a stipulated interval of 2 hours. Subsequent to this treatment, pH normalization to 7 was meticulously executed through the judicious addition of 1 M NaOH. The processed samples were then subjected to SDS-PAGE for further rigorous analysis with UPLC-HRMS.

Flow cytometry: Surface expression of TfR1 in HEK293T, A549, U87, MDCK and CHO was analyzed by staining with transferrin receptor 1 mouse antibody (FITC) (11020-R040-F, Sino Biological) at room temperature for 1 hour. Cells were washed with PBST twice and resuspended with PBS containing 1% FBS. Samples were analyzed with FlowJo V10 and CytExpert (version 2.4).

Western Blotting: The cells were lysed with RIPA lysate (adding protease inhibitor previously) and placed on ice for 15min, followed by centrifugation of 14, 000 × g, 20 min. The total protein was quantified using BCA assay to normalize. After sample loading, protein electrophoresis was performed using Tris-MOPS-SDS running buffer (M00138, Genscript) at 140V for 1 hour. Then protein was wet-transferred onto PVDF membrane at 100V for 1 hour with ice-water bath. The membrane was blocked with 5% milk in TBST, incubated with specific primary antibody at 4°C overnight and HRP-conjugated secondary antibody at room temperature for 1 hour. After washing with TBST for five times, the immunoblotting bands were exposed by enhanced chemiluminescence (ECL). Afterwards, the membrane was stripped and re-incubated with mouse monoclonal antibody against GAPDH to confirm the equal sample loading.

RNA isolation and quantitative RT–PCR analysis: The total RNA was extracted using TaKaRa MiniBEST Universal RNA Extraction Kit (9767, Takara). For viral RNA quantification. 2 μg of total RNA was reverse-transcribed using Hifair II 1st Strand cDNA Synthesis SuperMix for qPCR (11123ES60, Yeasen). The viral RNA level was measured by RT-qPCR using SYBR Green GoTaq qPCR Master Mix (A6001, Promega) on Anglient Mx3000P.

Primers used are as follows: for influenza A/WSN/33 M1 (forward: 5' GACCAATCC-TGTCACCTC 3' and reverse: 5' GATCTCCGTTCCCATTAAGAG 3'), for A/PR/8/1934 (H1N1) M1 (forward: 5' CCAATCCTGTCACCTCTGAC 3' and reverse: 5' TGGACAAAGCGTCTACGC 3'), for A/Aichi/2/1968 (H3N2) NP (forward: 5' CAAGAGTCAGCTGGTGTGGA 3' and reverse: 5' GCCCAGTACCTGCTTCTCAG 3'), for GAPDH (forward: 5' GAAGGTGAAGGTCGGAGTC 3' and reverse: 5' GAAGATGGTGATG-GGATTTC3'). The PCR conditions were 1 cycle at 95°C for 5min, followed by 40 cycles at 95°C for 15s, 60°C for 1min, and 1 cycle at 95°C for 15s, 60°C for 15s, 95°C for 15s.

RNA Interference: The siRNA against TfR1 and non-targeting siRNA were designed and synthesized from Hanbio (China). siRNA targeting TfR1: 5'-GCUGGUCAGUUCGUGAUUAdTdT-3', siRNA negative

ctrl: 5'-UUCUCCGAACGUGUCACGUdTdT-3'. $1 \times 10^5$ of A549 cells per well were seeded to 12-well plate overnight, followed by transfecting TfR1 and non-targeting siRNA (50 nM) for 48 hours which was delivered by Lipofectamine RNAi-MAX Transfection Reagent (13778030, Thermo) with the manufacturer's instructions. Successful siRNA knockdown was confirmed by western blotting.

CRISPR-Cas9 knockout: The oligonucleotide sequence is listed as follows: TfR1-sgRNA1: CTGAACGGGGT-ATATGACAA; TfR1-sgRNA2: CAGGAACCGAGTCTCCAGTG; TfR1-sgRNA3: AAATTCATATGTCCCTCGTG. sgRNAs were transcribed and screened with Guide-it sgRNA In Vitro Transcription Kit (632635, Clontech) and Guide-it Complete sgRNA Screening System (632636, Clontech). sgRNAs were cloned into LentiCRISPRv2 plasmid. LentiCRISPRv2 plasmid was co-transfected with psPAX2 and GFP plasmid to package Crispr-Cas9-lentivirus. A549 cells were infected with Crispr-Cas9-lentivirus for 48 hours and then change medium (DMEM with 1.5 µg/mL puromycin) for 14 days to select transduced cells. Successful CRISPR-Cas9 knockout was confirmed by western blotting.

Plasmid overexpression: 293T cells were seeded into 12-well plates at a density of $1 \times 10^5$ cells per well 24 hours before transfection. 1 µg expression plasmid (TfR1, SLC3A2 or other target proteins) with transfection reagent PEI were mixed and transfected in 293T cells. The expression of target proteins were quantified through western blotting.

Immunofluorescence: A549 cells ($2 \times 10^5$) were seeded into φ15mm confocal laser dish (801002, NEST) overnight. Cells were infected with influenza virus (MOI = 0.5) at indicated time and temperature. After briefly washing with PBS three times, cells were fixed with 4% paraformaldeh-yde in PBST for 20 min, permeabilized with 0.5% Triton X-100 in PBST for 20 min and blocked with 5% BSA in PBST for 1 hour at room temperature. Cells were incubated with primary mouse monoclonal antibody against TfR1 or influenza NP at 4°C overnight. Then cells were incubated with Alexa Fluor 488/647-conjugated goat anti-mouse IgG secondary antibody for 1 hour at room temperature. After that, cells were washed with PBST twice and stained with DAPI. Images were captured using Carl Zeiss LSM880 fluorescence micro-scope with a 63X oil immersion objective.

Measurement of viral infection: A549 cells were seeded into 12-well plate at the density of $1 \times 10^5$ cells per well over-night. To measure virus infection, cells were infected with IAV (MOI = 0.5) at 37°C for 12 hours to allow virus complete a full round of infection. Afterwards, the total RNA or protein was extracted to quantify viral infection levels. To measure virus entry, cells were infected with IAV (MOI = 5, increasing the virus MOI to detect viruses during transient infections) at 4°C for 1 hour and then transferred to 37°C for 1 hour. Cells were washed with PBS twice and then incubated with 50 ng/mL Proteinase K for 3 minutes to remove surface bound virus. Cells were centrifuged and washed with PBS twice and total RNA was extracted.

Measurement of viral attachment, internalization and replication: A549 cells were seeded at the density of $2 \times 10^5$ cells per well for 12 hours. To measure virus attachment, cells were washed with prechilled PBS and then incubated with influenza virus (MOI = 10) on ice for 2 hours to allow virus attachment. After that, cells were washed with chilled PBS to remove unbound virus. Then total RNA was extracted and virus genome was quantified with reverse transcription-quantitative PCR (RT-qPCR). The assessment of viral attachment requires termination of the experiment at early infection stages to avoid interference from subsequent processes such as endocytosis and membrane fusion. qRT-PCR directly quantifies viral genomic RNA (vRNA) adsorbed to the cell surface, leveraging its high sensitivity and rapid quantification capabilities, making it more optimal for precise detection at short time points compared to plaque assays.

To measure virus internalization, the concentrated IAV stock solution was diluted to 100 HA units, and labelled with suflo-NHS-SS-biotin (Thermo Fisher) at 1 µM for 2 hours at room temperature (with or without 0.1% NaN$_3$). The reaction was terminated by adding 0.1 M glycine (pH 7). The labelled virus was purified by ultrafiltration with 30 kDa ultrafiltration tube. $2 \times 10^5$ A549 cells were washed with PBS, treated with 0.05% trypsin, centrifuged at 200 g for 2 min, then resus-pended in 200 µL DMEM containing 1% FBS. IAV- SS-biotin (MOI = 5) was bound to host cells for 1 hour, then cells were centrifuged at 200 g for 1 min and resuspended in 200 µL DMEM containing 1% FBS and placed at 37°C for 1 hour. 150 µL of frozen 15 mM Tris (2-carboxyethyl) phosphine (TCEP) was added for 5 min, the cells were centrifuged at 200 g for

2 min and the cell pellets were resuspended in 4% PFA for 15 min. The cells were permeabilized with 0.5% Trion-X100, stained with 1 µg/ml Alexa Fluor 488 conjugated-Streptavidin (35103ES60, Yeasen) for 30 min, and the fluorescence intensity was measured by flow cytometry.

To determine virus replication, cells were incubated with 10 mM $NH_4Cl$ in DMEM at pH 7.4. (inhibiting virus maturation and further rounds of infection) for 1 hour before IAV 12-plasmids transfection. 8 hours after transfection, medium was discarded and cells were washed with PBS twice. Viral RNA was extracted and quantified by RT-qPCR.

Enzyme linked immunosorbent assay: ELISA Microplate (448496, Thermo) was coated with 500 ng TfR1 protein per well at 4°C overnight. Then microplate was blocked with 5% BSA in PBST for 1 hour at room temperature and then incubated with various concentrations of purified influenza virus for 2 hours at room temperature. After washing with PBST twice, plate was incubated with primary mouse monoclonal antibody against influenza virus HA for 2 hours and HRP-conjugated goat anti-mouse IgG secondary antibody for 1 hour at room temperature. Substrate tetramethylbenzidine was added after washing with PBST, followed by quenching with stop solution. Absorbance was measured at a wavelength of 450nm using Microplate Reader (Tecan Infinite M2000 PRO; Tecan Group Ltd., Switzerland).

Docking analysis: Protein structures of influenza virus neuraminidase (NA; PDB ID: 6Q23) and transferrin receptor 1 (TfR1; PDB ID: 1SUV) were retrieved from the Protein Data Bank (https://www.rcsb.org/). Molecular docking simulations were performed using the HDOCK server (http://hdock.phys.hust.edu.cn/), a web-based platform for protein-protein interaction prediction. The docking protocol utilized default parameters to generate an ensemble of putative binding conformations. The most suitable conformation, was selected for further structural analysis. Protein-protein interaction was visualized using PyMOL to identify binding conformation involved in the NA-TfR1 interaction.

Cell viability assay: A549 cells were seeded into 96-well plate at a density of $2 \times 10^4$ cells per well overnight. Cells were transfected with non-targeting and TfR1-specific siRNA (50 nM) for 48 hours. Cell viability was measured using Cell Titer-Glo reagent (G7570, Promega). The buffer was mixed with substrate to dissolve and added 40 µL per well into 96-well plate. Mix the contents for 2 min on an orbital shaker to induce cell lysis, allowing the plate to incubate at room temperature for 10 min to stabilize luminescent signal and luminescence was read using Microplate Reader.

Surface Plasmon Resonance (SPR): SPR binding experiments were performed using Biacore 8K (2383429, GE Healthcare) at room temperature to measure the kinetics and affinity of transferrin receptor 1 binding to influenza virus. Protein and virus solution were exchanged to PBS containing 0.05% (v/v) Tween-20 using Zeba Spin Desalting Columns (89882, Thermo). Purified Influenza A Virus (A/WSN/1933) or viral proteins were immobilized on CM5 chips respectively with standard amine-coupling procedure to about 3000 units (RU). Various concentrations of TfR1 protein (18.6 nM, 56.6 nM, 167 nM, 500 nM and 1500 nM) prepared from stock solution were flowed over sensor chip surface at a flow rate of 10 µL/min to perform single-cycle experiments. The binding kinetics and affinity constants were calculated using BIACORE INSIGHT EVALUATION software and the curve was fitted with a 1:1 binding model.

Co-immunoprecipitation: A549 cells were seeded to 6 cm dishes and incubated at 37°C for 12 hours. Cells were then infected with influenza virus (MOI = 20) for 2 or 6 hours at 37°C. Then medium was removed and cells were washed with modified PBS twice. Cells were harvested and lysed with IP lysis buffer on ice, followed by centrifugation (14, 000 × g, 15 min) to obtain supernatant. The supernatant was incubated with anti-TfR1 or mouse IgG agarose and rotated at 4°C overnight. The agarose was washed with IP lysis buffer for three times and then eluted with elute buffer. The eluate was boiling with SDS-PAGE loading buffer for western blotting.

*Ligand Competition with influenza virus*: A549 cells were seed into 12-well plate at a density of $2 \times 10^5$ cells per well overnight. Transferrin (Tf) protein was used to compete with IAV (MOI = 0.5) to bind host cells. 2 mg/mL Tf was added at indicated time points (2 hours before IAV infection, the same time with IAV infection or 2 hours after IAV infection). Virus RNA level was extracted 3 h after infection and quantified by RT-qPCR.

Antibody/protein inhibition of influenza virus: To determine whether TfR1 antibody could block IAV infection, A549 cells were seed into 12-well plate at a density of $2 \times 10^5$ cells per well overnight, followed by incubated with 20 µg/mL mouse

TfR1 monoclonal antibody at 37°C for 1 hour. Cells were infected with IAV (MOI = 0.5) at 4°C for 2 hours and then washed with PBS twice. Prewarmed medium was added and cells were shifted to 37°C for 4 hours. Afterwards, viral RNA was extracted and quantified by RT-qPCR. To determine whether recombinant TfR1 (ECD) protein could block IAV infection, A549 cells were seed into 12-well plate at a density of $2 \times 10^5$ cells per well overnight. IAV (MOI = 0.5) was incubated with 50 µg/mL TfR1$^{ECD}$ at room temperature for 1 hour and then infected A549 cells for 6 hours. Afterwards, viral RNA was extracted and quantified by RT-qPCR

Statistical analysis: Data were presented as mean ± SD. Statistical analysis was performed using GraphPad Prism (GraphPad Software, Version 9.0). Statistical significance was calculated using a two-tailed Student's t test and p value < 0.05 was considered significant. Significance is noted with asterisks as described in the figure legends. Animal experiments were not blinded or randomized, and no animals or samples were removed as outliers from the analysis.

## Supporting information

**S1 Fig. Validation of photo-crosslinking conjugation in IAV particles with a specific focus on TfR1 as the primary entry receptor.** (A) Diagram illustrating the capture of factors interacting with influenza virus using photo-crosslinking and UPLC-MS. Azido-displayed IAV (IAV-N$_3$) was created by converting Ac$_4$ManNAz into terminal sialic acid, and the photo-crosslinking probe was conjugated through a click reaction. TfR1 was captured through UV irradiation, followed by streptavidin enrichment and subsequent identification in proteome analysis. Schematic created using BioRender (https://Biorender.com) (B) Modification of IAV envelope proteins using DIBO-DAZ-Biotin probe did not display any discernible impact on IAV infectivity. A549 cells were infected with probe labelled virus for 12 hours and viral infectivity was confirmed by RT-qPCR. (C) Western blotting characterization of the effect of TfR1 overexpression in Lec1 cells on IAV infection (MOI = 0.5). Equal loading was confirmed by detecting GAPDH. (D) Colocalization analysis of TfR1 with influenza virus envelope proteins HA/NA during IAV endocytosis. A549 cells were infected with IAV (MOI = 0.5) for 2 hours, then fixed and stained with viral antibodies. The Pearson's Coefficient between TfR1 and HA/NA was 0.928/ 0.802. HA/NA in red, TfR1 in green, nuclei in blue. Scale bars, 50 µm. (E) Colocalization analysis of TfR1/SLC3A2 with influenza virus envelope proteins NA during IAV endocytosis. A549 cells were infected with IAV (MOI = 5) for 2 hours, then fixed and stained with viral antibodies. TfR1/SLC3A2 in red, NA in green, nuclei in blue. Scale bars, 10 µm. Unpaired t-tests were used for statistical analysis, denoting significance as follows: *p < 0.05; **p < 0.01; ***p < 0.001; and ****p < 0.0001, while "n.s." indicates non-significance.
(TIFF)

**S2 Fig. Inhibiting IAV infection through targeted disruption of TfR1 role in IAV entry.** (A) Comparison of sialic acid and TfR1 as potential receptors in mediating IAV entry. A549 cells were transfected with TfR1/vector plasmid or TfR1 siRNA, treated with neuraminidase/BSA at 37°C for 2 hours, and infected by IAVs (MOI = 0.5), which were pre-treated with neuraminidase or untreated. Additionally, influenza virus (MOI = 0.5) was treated with neuraminidase/BSA in glycol buffer at 37°C for 2 hours before adding to A549 cells. Immunofluorescence was used to visualize IAV infection, with NP in green, nuclei in blue. Scale bars, 50 µm. (B) Flow cytometry analysis of surface TfR1 expression and its correlation with the promotion of IAV endocytosis in wild-type, TfR1 knockout, TfR1 rescued A549 cells, and CHO cells. Mean fluorescence intensity was used for further quantification. (C) Assessment of TfR1's involvement in initiating the clathrin-mediated pathway (CMP) in IAV endocytosis. A549 cells were either transfected with siRNA targeting TTP or CLTA or treated with various endocytosis inhibitors, followed by infection with IAVs (MOI = 0.5). Subsequent quantification was conducted using western blotting or RT-qPCR. Equal loading was confirmed by detecting GAPDH. Unpaired t-tests were used for statistical analysis, denoting significance as follows: *p < 0.05; **p < 0.01; ***p < 0.001; and ****p < 0.0001, while "n.s." indicates non-significance. (D) Visual depiction of the further colocalization of NA and TfR1 (seen S1D Fig) with downstream essential factors (TTP/CLTA) in the clathrin-mediated pathway during IAV endocytosis. NA in blue, TfR1 in green, TTP/

CLTA in red, Nuclei in cyan. This colocalization, consistent with data in S1D Fig, provides direct visual evidence for the role of TfR1 in mediating IAV endocytosis. Scale bars: 50 μm. Pearson's coefficients (calculated from zoomed-in regions): NA-TfR1 = 0.672, NA-TTP = 0.689, NA-CLTA = 0.701, TfR1-CLTA = 0.744. (E) Comparative evaluation of sialic acid versus TfR1 as potential receptors in mediating IAV entry. Lec1 cells were transfected with TfR1/vector plasmid 48 hours before IAV infection (MOI = 0.5). Infected cells were lysed and quantified by RT-qPCR after 12 hours. (F) Assessment of the inhibitory effect of TfR1 knockdown on A/WSN/1933 (H1N1)/VSV infection (MOI = 0.5) in A549 cells. Infected cells were lysed and quantified by RT-qPCR after 2,4,6,12,24 hours. (G) Flow cytometry gating strategies employed to TfR1 involvement in IAV entry(Fig 2F and 2G). Unpaired t-tests were performed for statistical analysis, with significance denoted as follows: *$p < 0.05$; **$p < 0.01$; and ***$p < 0.001$, while "n.s." indicates non-significance.
(TIFF)

**S3 Fig. Unraveling the mechanism underlying TfR1-mediated IAV entry.** (A) Modulation of TfR1 expression through treatment with various small molecule compounds either restricted or promoted IAV infection. A549 cells were treated with Ferric ammonium citrate/Deferoxamine/Ferristatin-II (100 μM) or an equivalent volume of DMSO for 24 hours, followed by infection with influenza virus (MOI = 0.5). Viral M1 protein was quantified using western blotting or RT-qPCR. Equal loading was confirmed by detecting GAPDH. (B) Evaluation of a small molecule degrader targeting TfR1 in a mouse model. BALB/c mice were divided into three groups (5 mice/group). Mice were intraperitoneally administered Ferristatin-II (20mg/kg) or vehicle (1% DMSO in PBS) daily for 5 days, then challenged with 5x $LD_{50}$ A/WSN/1933 (H1N1). The survival rate of the mice was monitored and recorded. (C) Disruption of the TfR1-IAV interaction using anti-TfR1 antibody or TfR1-ectodomain protein significantly impaired IAV infection (MOI = 0.5), contrast to TfR1's ligand transferrin (targeting the helical domain) (2mg/mL) for 6 hours. The virus titer was quantified using RT-qPCR. (D) Neither the binding of influenza virus nor the antibody targeting the TfR1 apical domain (OKT-9) interfered with its ligand transferrin uptake. A549 cells were incubated with AF568 labelled transferrin in the presence of OKT-9 or IAV (MOI = 0.5) for 6 hours, transferrin uptake was quantified via flow cytometry. (E) Illustration demonstrating different extracellular truncated TfR1 variants (distinctly colored) and their respective associations with facilitating IAV infection in CHO cells. Various truncated TfR1 plasmids were transfected into CHO cells 48 hours before IAV infection (MOI = 0.5). Virus infectivity was determined by RT-qPCR and compared with vector. (F) Identification of viral components interacting with TfR1 through co-immunoprecipitation assay and western blotting. Lysates from A549 cells at 6 hours post-infection with IAV (MOI = 20) were co-immunoprecipitated using anti-TfR1 antibody along with anti-viral antibodies. Unpaired t-tests were conducted for statistical analysis, and significance levels were indicated as follows: *$p < 0.05$; **$p < 0.01$; ***$p < 0.001$; and ****$p < 0.0001$, while "n.s." denoted non-significance.
(TIFF)

**S4 Fig. Exploration of the interaction between TfR1 and NA, constructing a comprehensive interacting model.**
(A) Identification of photo-crosslinked sequences and specific crosslinked sites within TfR1 was achieved using high-resolution mass spectrometry (HRMS). TfR1 protein was incubated with a cleavable photo-crosslinking probe labeled with the mutant IAV at NA-N44A/N72A. Following UV irradiation and SDS-PAGE separation, the sample underwent UPLC-HRMS for further analysis. (B) Characterization of the key glycosylation sites on NA interacting with TfR1 was performed using entry assays and ELISA. TfR1 was immobilized on plates prior to the assays. (C) OKT-9 antibody attenuated IAV infection by specifically binding to the apical domain of TfR1. A549 cells were infected with NHS-SS-biotin labelled IAV (MOI = 0.5) in the presence of OKT-9 or IgG ctrl for 6 hours and then IAV particles uptake was quantified via flow cytometry. (D) Characterization of TfR1's affinity with IAV NA WT/mutant proteins using Surface Plasmon Resonance (SPR) experiments. NA WT/ mutant proteins were immobilized on CM5 chips and interacted with TfR1 protein to measure the binding affinity constants. Unpaired t-tests were conducted for statistical analysis, and significance levels were indicated as follows: *$p < 0.05$; **$p < 0.01$; ***$p < 0.001$; and ****$p < 0.0001$, while "n.s." denoted non-significance.
(TIFF)

**S5 Fig. Identification of TfR1 region responsible for triggering matrix protein degradation.** (A) Prolonged exposure to influenza virus (MOI = 0.1) did not exhibit an impact on TfR1 mRNA levels, as assessed through RT-qPCR. (B) Assessment of the impact of viral M1 protein on TfR1 by co-transfecting the TfR1 plasmid (0.5 µg) with varying concentrations of M1 or PB2 plasmid in 293T cells for 48 hours. Equal loading was confirmed by detecting GAPDH. (C) Investigation into the involvement of aggresome systems in TfR1-mediated degradation of the M1 protein. 293T cells were co-transfected with the TfR1 plasmid and viral plasmids expressing M1 or NP in the presence or absence of Tubacin (5 µM). Protein levels of TfR1, M1, and NP were assessed by western blotting 48 hours post-transfection. Equal loading was confirmed by detecting GAPDH. (D) Investigation into the involvement of aggresome systems in virus infection. A549 cells were infected with IAV (MOI = 0.1) in the presence or absence of Tubacin (5 µM) for 12 hours and viral infectivity was confirmed by RT-qPCR. (E) Examination of the interplay between lysosomal and proteasomal systems in TfR1-driven degradation of M1 protein. 293T cells were co-transfected with the M1 expression plasmid and TfR1 expression plasmid, in the presence or absence of HCQ (50 µM, 250 µM) or MG-132 (10 µM, 50 µM). Protein levels of TfR1 and M1 were evaluated by western blotting 48hours post-transfection. Equal loading was confirmed by detecting GAPDH. (F) Western blotting analysis to determine the specificity of TfR1-mediated viral protein degradation. Individual plasmids from the IAV plasmid complex were co-transfected with the TfR1 plasmid for 48 hours, employing the SLC3A2 expression plasmid as a control background. Equal loading was confirmed by detecting GAPDH. (G) RT-qPCR analysis to exclude plasmid competition TfR1-mediated viral protein downregulation. 293T cells were co-transfected with the TfR1/SLC3A2/Vector plasmid for 48 hours. The M1 RNA level was was confirmed by RT-qPCR (H) The critical role of the intracellular domain of TfR1 in viral M1 degradation. Various TfR1 and SLC3A2 expression plasmids with interchanged intra-/extracellular domains were transfected along with the 12 IAV packaging plasmids complex. Cellular state images were captured three days post-transfection. (I) Visual depiction of the TfR1 transmembrane domain (red) and intracellular domain (blue) structure alongside various TfR1 truncations. Statistical analysis was conducted using unpaired t-tests, and the significance levels were denoted as follows: *$p < 0.05$; **$p < 0.01$; ***$p < 0.001$; and ****$p < 0.0001$, while "n.s." represented non-significance. (TIFF)

**S6 Fig. The translational potentials of TfR1-M1 degradation hold promise as a novel avenue for antiviral therapeutics.** (A) The distinct effects of various TfR1 truncations on IAV packaging were observed. 293T cells were transfected with the 12 IAV package plasmids complex and different TfR1 expression plasmid fragments. Cell state images were captured 3 days after transfection. The supernatant containing IAVs was lysed and subsequently analyzed using RT-qPCR or western blotting. Equal loading was confirmed by detecting GAPDH. (B) Demonstrating the substantial antiviral effect of dimerized truncated TfR1. 293T cells were transfected with expression plasmids for truncated TfR1 containing an isoleucine zipper or GST as control, followed by influenza virus infection (MOI = 0.1). Visual representations of resulting cell cytopathic effects (CPE) were captured in images. Protein levels of M1 were assessed via western blotting 48 hours post-transfection. Equal loading was confirmed by detecting GAPDH. Immunofluorescent staining depicting 293T cells transfected with TfR1 or truncated TfR1, TfR1 or truncated TfR1 (violet), Nuclei (DAPI, blue). (C) Illustrating the dose-dependent degradation of RABV-matrix and MeV-matrix proteins by TfR1. Co-transfection involved RABV-matrix/MeV-matrix expression plasmids with escalating concentrations of TfR1 plasmid in 293T cells, followed by analysis through western blotting 48 hours after transfection. Equal loading was confirmed by detecting GAPDH. (D) Assessment of the effect of TfR1 truncation on virus entry by various intracellularly truncated versions of TfR1. CHO cells were transfected with truncated TfR1 plasmids, followed by influenza virus infection (MOI = 0.5). Infected cells were lysed and quantified by RT-qPCR after 6 hours. (E) Subcellular localization of truncated TfR1 mutants. 293T cells were transfected with expression plasmids encoding full-length TfR1 or truncated TfR1 variants (ΔTM and Δ1–67). Cells were fixed and subjected to immunofluorescence staining, TfR1 or truncated TfR1(Green), Nuclei (DAPI, blue). Unpaired t-tests were performed for statistical analysis, with significance denoted as follows: *$p < 0.05$; **$p < 0.01$ and ***$p < 0.001$, while "n.s." indicates non-significance. (PNG)

**S1 Movie. (separate file). Docking analysis of NA tetramer (6q23) and TfR1 (1suv) dimer.** The proposed binding sites of NA occupy the apical domain of TfR1, considering a 3 Å resolution X-ray structure. The NA tetramer binds to the apical domain of the TfR1 dimer, proximal to the crosslinked peptide of TfR1. (MP4)

## Acknowledgments

We would like to thank J. Wang from the State Key Laboratory of Natural and Biomimetic Drugs for support with Biacore Surface Plasmon Resonance.

## Author contributions

**Conceptualization:** Yuanhao Li, Xinchen Wang, Dezhong Ji, Yuanjie Zhang, Zhiqian Chen, Qiuchen He, Chuanling Zhang, Sulong Xiao, Lihe Zhang, Demin Zhou.

**Data curation:** Yuanhao Li, Xinchen Wang, Yiming Wang, Xiaoyang Wang, Kangming Guo, Yu Mu, Chen Qin, Tao Yuan, Zhiqian Chen, Xingxing Zhu, Honghui Jiang.

**Formal analysis:** Yuanhao Li, Xinchen Wang.

**Funding acquisition:** Dezhong Ji, Tao Yuan, Honghui Jiang.

**Investigation:** Yuanhao Li.

**Methodology:** Yuanhao Li, Xinchen Wang, Mengyang Wang, Xiaohui Zhang, Chuanling Zhang.

**Validation:** Yuanjie Zhang.

**Writing – original draft:** Yuanhao Li, Xinchen Wang, Dezhong Ji, Demin Zhou.

**Writing – review & editing:** Yuanhao Li, Dezhong Ji, Demin Zhou.

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
