## [Decision Letter · Decision Letter 0]

2 Nov 2024

PPATHOGENS-D-24-01941TfR1 facilitates influenza virus endocytosis and uncoating by interacting with NA and M1 via extracellular and intracellular domainsPLOS Pathogens Dear Dr. Zhou, Thank you for submitting your manuscript to PLOS Pathogens. After careful consideration, we feel that it has merit but does not fully meet PLOS Pathogens's publication criteria as it currently stands. Therefore, we invite you to submit a revised version of the manuscript that addresses the points raised during the review process. Please submit your revised manuscript within 60 days, that is the end of the year. If you will need more time than this to complete your revisions, please reply to this message or contact the journal office at plospathogens@plos.org. Please include the following items when submitting your revised manuscript:* A rebuttal letter that responds to each point raised by the editor and reviewer(s). You should upload this letter as a separate file labeled 'Response to Reviewers '. This file does not need to include responses to any formatting updates and technical items listed in the 'Journal Requirements' section below.* A marked-up copy of your manuscript that highlights changes made to the original version. You should upload this as a separate file labeled 'Revised Manuscript with Track Changes '.* An unmarked version of your revised paper without tracked changes. You should upload this as a separate file labeled 'Manuscript '. If you would like to make changes to your financial disclosure, competing interests statement, or data availability statement, please make these updates within the submission form at the time of resubmission. Guidelines for resubmitting your figure files are available below the reviewer comments at the end of this letter. We look forward to receiving your revised manuscript. Kind regards, Maria João Amorim, Ph.DAcademic EditorPLOS Pathogens Matthias SchnellSection EditorPLOS Pathogens Michael Malim

Editor-in-Chief

PLOS Pathogens

orcid.org/0000-0002-7699-2064   **Journal Requirements:** **Additional Editor Comments (if provided):** Dear Dr. Zhou,

Thank you for your patience while your manuscript "TfR1 facilitates influenza virus endocytosis and uncoating by interacting with NA and M1 via extracellular and intracellular domains" was peer-reviewed at PLOS Pathogens. Your study has now been assessed by reviewers with relevant expertise. As you will see in their reports at the end of this email, despite noting the interest of the study, the reviewers raise significant concerns about the conclusiveness of the work.

Based on these reviews, I regret to inform you that we may only consider an invited substantial revised version of the study. We are aware that successfully addressing the concerns raised will entail a significant amount of work and that the current conclusions may not be supported, but we will try to engage the same academic editor and reviewers to evaluate the work. The issues raised regarding overstatements in the manuscript and the need to re-write parts of the manuscript would need to be addressed.

Also, the reviewers raise serious concerns about the data including:

- How can Trf1 be essential for infection but also act as an antiviral?;

- The authors used a late time point of infection rather than an early time point to observe viral entry and hence the experiments should be repeated at an earlier time point;

- There are many control experiments missing including on viral entry and on the use of treatments such as Tubacin.

- The experiments should be repeated with different IAV strains,

- The quality of figures and explanations needs to be improved.

Because producing the required data would require substantial cost and effort with an uncertain outcome, we understand that you may prefer to pursue faster publication of this work elsewhere.

I am sorry that we cannot be more positive on this occasion. I thank you for having considered PLOS Pathogens for publication and hope that you will find the reviewers' feedback helpful as you consider how to proceed with this work.**Reviewers' Comments:** Reviewer's Responses to Questions

**Part I - Summary**

Reviewer #1: The authors use a novel virus labelling technique to study host cell receptors of IAV. This allowed them to identify transferrin reporter 1 as a putative IAV receptor which has been previously described by Mazel-Sanchez et al. 2023 (PNAS). While the latter paper describes that HA interacts with TfR1 in a sialic acid-dependent manner, here the authors claim that it is not HA but NA which interacts with TfR1, facilitating virus entry. In addition, the authors claim that TfR1 interacts with the matrix protein M1 and aids in viral uncoating by targeting M1 for proteasomal degradation. The authors have clearly put a lot of effort into compiling a significant amount of data in this manuscript. However, despite their efforts, the experiments and analyses presented need further refinement to support the claims made, and several critical issues in experimental design, data interpretation, and presentation should be addressed.

Reviewer #2: (No Response)

Reviewer #3: Strengths: molecular analysis of the TfR1-Neuramindase interaction and its importance in IAV entry.

Weaknesses: virus entry experiments and aggresome pathway experiments

Novelty: similar to the strengths and promising in vivo data.

Execution: the authors are on the right track regarding the TfR1-NA interaction, but some experiments are inadequate for the purpose, or over-interpreted.

**Part II – Major Issues: Key Experiments Required for Acceptance**

Reviewer #1: 1. Figure 2F: A control experiment is missing to show that the assay can discriminate between attached and internalized virus. After 2 hours at 37°C, some virus will remain attached to the cell surface. The authors should include controls such as endocytosis inhibitors or biotinylated virus with and without permeabilization to ensure that they are measuring only internalized virus (see PMID: 26575457 for relevant methods).

2. Figure 3A: The co-immunoprecipitations are performed after 24 hours of infection, but it would be more informative to perform these experiments shortly after cold binding or early post-infection (e.g., 1-2 hours post-infection) at a high MOI (e.g. MOI=10 or 20) to test for interactions. This applies to the entire manuscript.

3. Figure S4B: The loss of binding to TfR1 by the NA mutant N219A should be confirmed using surface plasmon resonance (SPR). In addition, virus entry studies should be performed with a recombinant virus harboring the N219A mutation in NA, to confirm its role in TfR1-mediated entry and uncoating.

4. Figure 5B: Expression controls for the chimeric constructs are missing and should be included.

5. Figure 5C: The effect of TfR1 truncation on virus entry should be investigated and discussed.

6. Figure 6D: Conflicting and confusing results with data presented in previous Figures where TfR1 promotes IAV infection, here TfR1 inhibits IAV replication in a dose-dependent manner. Virus growth curves on wild type, TfR1 KO and TfR1 over expressing A549 or 293T cells should be performed.

7. Figure S6: Localization of TfR1 truncation mutants need to be assessed by immunofluorescence or flow cytometry. Negative controls with expression plasmids (e.g. GST) need to be added to show specificity of the observed effects on M degradation.

Reviewer #2: (No Response)

Reviewer #3: 1. Results section 2.4: It is unclear what the role of HDAC6 is in the TfR1-mediated degradation of M1. During aggresome formation, proteasome activity is downregulated, which explains the inhibition of M1 degradation. Additionally, Trf1 and M1 co-localize with HDAC6 in the aggresome. However, based on our experience, when a protein is overexpressed, it typically aggregates in the aggresome after proteasome inhibition. The confusion lies in the application of Tubacin. Tubacin only interferes with HDAC6’s enzymatic activity, not its ubiquitin-binding ability, which regulates aggresomes. How can the authors conclude that the HDAC6-Ub pathway is affected? Furthermore, while Tubacin inhibits aggresome formation, this does not logically connect to the subsequent data (Fig. 4F to H). In Fig. 4G, the green aggregates are not aggresomes. Aggresomes should be unique within the cell. Furthermore, the figure legends are vague. What do the arrows indicate? What does the cyan color represent? These details should be clearly explained.

2. In Fig. 2A, the knockdown of TfR1 inhibits IAV infection, indicating that TfR1 is essential for viral infection, which seems to suggest a proviral role. However, at the end of the results section, the authors claim that TfR1 acts as an antiviral agent by degrading the M1 protein. What is the exact role of TfR1? Is the extracellular domain necessary for virus infection while the intracellular domain is responsible for antiviral M1 degradation? Additionally, Fig. 1D shows that high levels of TfR1 correspond to high levels of M1 and increased IAV susceptibility. Does this contradict the later finding that high TfR1 levels lead to M1 degradation? That the authors use a 24 hpi time point to co-IP the TFr1 and M1 is confusing. 24 hpi is a very late time point of infection and has little to do with IAV entry, but more with assembly and release.

3. Virus internalization assays (Page 34) were done in cell suspension with DMEM + 1% FCS. FCS is blocks IAV binding and infectivity massively, and at extreme low concentrations blocks CME, re-routes the virus to a dynamin-independent macropinocytic pathway (de Vries et al., 2023, was not cited). The experiments containing NH4Cl were not appeared to be buffered at pH 7.4 with HEPES (it is not clarified in the methods). How does 0.05% Trypsin affect surface TfR1 levels?

See De Vries et al., (2011) Plos Path https://doi.org/10.1371/journal.ppat.1001329 Dissection of the Influenza A Virus Endocytic Routes Reveals Macropinocytosis as an Alternative Entry Pathway

The conclusion in Fig. 2F stating, “These findings suggest that TfR1 is more crucial for IAV endocytosis than for initial attachment to the cell surface. Consistently, further investigations into viral internalization were conducted using A549 cells,” is overinterpreted. Attachment and endocytosis are not independent steps; in fact, endocytosis follows attachment. Therefore, the observation of a greater inhibitory effect on endocytosis may be due to inefficient attachment, with the difference being amplified by the biological cascade.

**Part III – Minor Issues: Editorial and Data Presentation Modifications**

Reviewer #1: Figure 1A: The schematic showing IAV plasmids is confusing and does not add clarity to the overall understanding. Replacing this with an unlabeled illustration of an IAV particle would be more helpful.

Figure 1B: The multiplicity of infection (MOI) appears to be inaccurate throughout the manuscript. It appears to be closer to an MOI of 50, or it may indicate the presence of defective nuisance particles. The authors should perform a plaque assay comparing the tagged virus with the wild-type virus to determine if infectivity is affected by the tagging.

Figure 1C: The phrase "WSN package" is unclear. It would be clearer to simply state "WSN".

Text Clarity: The text states that WSN are "co-cultured" with A549 cells. This is incorrect and should be changed to "A549 cells were infected with WSN".

Figure 1 Panels: The order of panels D and E should be reversed for a more logical flow.

Figure 1D: The molecular weight ladder label is missing, and this oversight occurs in the Western blot (WB) images in the remaining figures. Please ensure that all WBs have properly labeled ladders for reference.

Figure 2B: The time post infection should be clearly indicated in this figure to provide context for the data presented.

Figure 2C: A virus growth curve should be included to complement the data. In addition, a negative control (e.g., vesicular stomatitis virus, VSV) should be included to demonstrate that the effect of TfR1 is specific to IAV.

Figure 2E: The graph labels are confusing, and the conditions should be clarified to make the data easier to interpret.

Figure 2G: The differentiation between A549 and CHO cells could be improved by using different colors or histogram styles to better illustrate the data.

Figure 2F: Lectin staining could also be used to show that TfR1 does not affect sialic acid levels.

Abbreviation: The abbreviation "CLTA" is not introduced in the manuscript and should be explained when first mentioned.

Figure 3A: Analyzed time points and MOI are not chosen appropriately to study IAV entry.

Figure 3B: The manuscript does not specify how the recombinant proteins (HA, NA, M1, M2, and NP) used for the binding studies were produced or purchased. This information should be clearly provided.

Figure 3C and 3D: The methodological explanations are insufficient to understand how the two TfR1 domains were selected as potential interacting partners with NA. This needs to be clarified.

Figure S3D: The term "negative control" is vague. It would be clearer to use "untreated" or "mock-treated" to describe the control group.

Figure S3E: The comparison for the statistical test is unclear. The authors need to specify which columns are being compared.

Figure S4B: Details on how the ELISAs were performed with the NA mutant viruses are missing. The authors should explain how the assays were normalized and confirm whether the viruses are viable. In addition, the Y-axis is labelled "% IAV infection," but it is unclear how the ELISA results were translated into infection rates.

Figure 4B: The experimental design is unclear. Increasing amounts of M1 and a negative IAV protein should be titrated with constant amounts of TfR1 to assess the interaction.

Figure 4G: The authors should include staining for both M1 and TfR1 in this figure.

Figure 4H: This figure needs a more detailed explanation of what was transfected and what was measured. The interaction between M1 and wild-type TfR1 is also missing and should be included.

Figure 5B: The choice of domain swap between SLC3A2 and TfR1 is questionable as it may not be the most relevant comparison. It would make more sense to swap domains with a related protein that does not affect M1 degradation. The authors might consider testing transferrin receptors from other species instead.

Figure 6: This figure is not convincing as presented and requires significant revision or additional data to support the claims.

Reviewer #2: (No Response)

Reviewer #3: 1. In Fig. 4B, the GAPDH levels are not consistent.

2. References; the main text cites Schmolke’s work (Mazel-Sanchez B et al., 2023) as [25], however, in the References it is [24].

3. Overinterpretation of the data.

4. End of the Introduction “This mechanism unveils the long-…” should be toned down.

5. For Fig. S1D and S2D, a quantitative analysis of co-localization should be performed.

6. Figure S2D is unfortunately not a convincing display of co-localization. For one thing, there is excessive NA signal (saturated) due to virus overload. Claims of co-localization require higher resolution to enable single virus resolution and performed at an optimal virus particle number per cell.

7. Fig. 2E is not adequately explained. Is TfR1 overexpressed? What is a Lec1 cell? Please clarify in the figure legend. What is the concentration of Neuramindase used?

8. In Fig. 3F, the interface analysis, particularly regarding hydrogen bonds based on docking simulations, is not recommended. Such information should be derived from the actual 3D structure, as predicted models cannot accurately indicate the positions of side chains. Additionally, the parameters and evaluations of the docking simulation should be provided.

9. In Fig. 4A, how did the authors maintain A549 cells in culture for 9 days?

PLOS authors have the option to publish the peer review history of their article (what does this mean? ). If published, this will include your full peer review and any attached files.

**Do you want your identity to be public for this peer review?** For information about this choice, including consent withdrawal, please see our Privacy Policy .

Reviewer #1: No

Reviewer #2: No

Reviewer #3: No

---

## [Decision Letter · Decision Letter 1]

23 Apr 2025

PPATHOGENS-D-24-01941R1

TfR1 facilitates influenza virus endocytosis and uncoating by interacting with NA and M1 via extracellular and intracellular domains

PLOS Pathogens

Dear Dr. Zhou,

Thank you for submitting your manuscript to PLOS Pathogens. After careful consideration, we feel that it has merit but does not fully meet PLOS Pathogens's publication criteria as it currently stands. Therefore, we invite you to submit a revised version of the manuscript that addresses the points raised during the review process.

In particular, the reviewers find the premise of the manuscript very interesting, but still need additional data and/or clarifications to be convinced that the observed structures are aggresomes. In particular,

- they request for inclusion of more controls to classify the structures as HDAC6/Ub/aggresomes. For example, including gamma tubulin could help to clarify whether these structures are found in close vicinity of MTOC.

- also the observed punctate staining pattern of M1 appears independent of TfR expression and typically, M1 expression would be expected to be diffuse. Could you please clarify?

Please submit your revised manuscript within 30 days Jun 22 2025 11:59PM. If you will need more time than this to complete your revisions, please reply to this message or contact the journal office at plospathogens@plos.org. Please include the following items when submitting your revised manuscript:

We look forward to receiving your revised manuscript.

Kind regards,

Maria João Amorim, Ph.D

Academic Editor

PLOS Pathogens

Matthias Schnell

Section Editor

PLOS Pathogens

Sumita Bhaduri-McIntosh

Editor-in-Chief

PLOS Pathogens

orcid.org/0000-0003-2946-9497

Michael Malim

Editor-in-Chief

PLOS Pathogens

orcid.org/0000-0002-7699-2064

**Additional Editor Comments :**

The reviewers find the premise of the manuscript very interesting, but still need additional data and/or clarifications to be convinced that the observed structures as aggresomes. In particular,

- they request for inclusion of more controls to classify the structures as HDAC6/Ub/aggresomes. For example, including gamma tubulin could help to clarify whether these structures are found in close vicinity of MTOC.

- also the observed punctate staining pattern of M1 appears independent of TfR expression and typically, M1 expression would be expected to be diffuse. Could you please clarify?

**Journal Requirements:**

1) Please confirm whether your study includes live participants. If so, please insert an Ethics Statement at the beginning of your Methods section, under a subheading 'Ethics Statement'. It must include:

i) The full name(s) of the Institutional Review Board(s) or Ethics Committee(s)

ii) The approval number(s), or a statement that approval was granted by the named board(s).

2) Some material included in your submission may be copyrighted. According to PLOSu2019s copyright policy, authors who use figures or other material (e.g., graphics, clipart, maps) from another author or copyright holder must demonstrate or obtain permission to publish this material under the Creative Commons Attribution 4.0 International (CC BY 4.0) License used by PLOS journals. Please closely review the details of PLOSu2019s copyright requirements here: PLOS Licenses and Copyright. If you need to request permissions from a copyright holder, you may use PLOS's Copyright Content Permission form.

Potential Copyright Issues:

i) Figure 6F. Please confirm whether you drew the images / clip-art within the figure panels by hand. If you did not draw the images, please provide (a) a link to the source of the images or icons and their license / terms of use; or (b) written permission from the copyright holder to publish the images or icons under our CC BY 4.0 license. Alternatively, you may replace the images with open source alternatives. See these open source resources you may use to replace images / clip-art:

3) We note that your Data Availability Statement is currently as follows: "All relevant data are within the manuscript and its Supporting Information files.". Please confirm at this time whether or not your submission contains all raw data required to replicate the results of your study. Authors must share the “minimal data set” for their submission. PLOS defines the minimal data set to consist of the data required to replicate all study findings reported in the article, as well as related metadata and methods (https://journals.plos.org/plosone/s/data-availability#loc-minimal-data-set-definition).

4) Please amend your detailed Financial Disclosure statement. This is published with the article. It must therefore be completed in full sentences and contain the exact wording you wish to be published.

5) Please ensure that the funders and grant numbers match between the Financial Disclosure field and the Funding Information tab in your submission form. Note that the funders must be provided in the same order in both places as well. Currently, the order of the grants is different in both places. In addition, these grants " 2017ZX09309009,91753202, and 2022A-157-G" are missing from the Funding Information tab. These grants "2024YFA0917500, and 2024A-167-G" are missing from the Financial Disclosure field.

6) The file inventory includes files for Figures S2-1, S2-2,  S3-1 and  S3-2. We would recommend either combining these into a single Figure S2, and Figure S3 file with separate internal panels, or renumbering them as individual figures, as we are not able to publish multiple components of a single figure as separate files.

**Reviewers' Comments:**

Reviewer's Responses to Questions

**Part I - Summary**

Reviewer #2: (No Response)

Reviewer #3: This article provides evidence that influenza A virus (IAV) glycoproteins interact with the host cell receptor TfR, facilitating virus entry. The authors further propose that the M1-TfR interaction promotes M1 degradation via the proteasome during viral escape from late endosomes. The biochemical data presented in the first half of the study, which identifies specific interaction sites on TfR, is interesting and well-supported. However, the second half of the study, exploring the proposed M1 degradation mechanism, relies largely on correlations and lacks some important controls, making these conclusions less robust.

**Part II – Major Issues: Key Experiments Required for Acceptance**

Reviewer #2: (No Response)

Reviewer #3: 1) HDAC6 and Tubacin experiments. Unfortunately, there are several shortcomings with the interpretation of HDAC6 and the aggresome pathway in TfR1-mediated M1 degradation. It is unclear why tubacin would prevent degradation of M1.Overexpression of the proteins under proteasome inhibition conditions commonly leads to protein aggregation. Therefore, the co-localization of ubiquitin with TfR1 and M1 shown in Fig. 4F provides weak evidence for the hypothesis proposed on p.27 that TfR1 induces misfolding of M1, subsequently directing it toward the ubiquitin-proteasome pathway. Could it be that ubiquitinated TfR1 directly targets M1 for degradation?

2) Proviral and antiviral roles of TfR1

The authors’ response indicates that TfR1-mediated degradation of M1 reduces progeny virus production, citing dose-dependent inhibition shown in Fig. 6D. However, co-transfection of multiple plasmids confounds the experiment - increasing the amount of one plasmid (TfR1) could affect transfection or expression efficiency of the other plasmids.

To support the authors’ conclusion, the following experiments are suggested:

• Include a suitable control experiment transfecting equal amounts of a neutral plasmid (e.g. SLC3A2 plasmid) to rule out non-specific competitive effects.

• Perform virus production assays using TfR1-knockdown or KO cells.

Additionally, the experiment shown in Fig. S2F measures viral genome replication during a single-cycle infection, which does not provide direct evidence of progeny virus production inhibition by M1 degradation. Genome replication and progeny virus production/release are distinct steps; thus, an experiment specifically assessing progeny virus titers is necessary to support author’s claims.

3) The role of TfR1 in IAV endocytosis

In Figure 2F, the flow cytometry data are convincing. However, for RT-qPCR-based quantifications it is unclear to the reviewer how they removed extracellular virus particles before measuring the genome of internalized viruses. If extracellular viruses were not removed, how did they distinguish between extra- and intra-cellular viruses?

**Part III – Minor Issues: Editorial and Data Presentation Modifications**

Reviewer #2: (No Response)

Reviewer #3: Minor issues (related to previous comments):

1) Figures S1D and S2D (co-localization analysis)

- Clearly describe the method used for quantifying colocalization (Pearson’s correlation coefficient). The observed high correlation is most likely due to using a low magnification for the confocal microscopy.

- They should add an additional control using an unrelated membrane protein such as SLC3A2 to validate the significance of the observed colocalization.

2) Neuraminidase concentration (Fig. 2E):

- Clarify the neuraminidase concentration used in units or μg/mL, not in dilution factor, for objectivity. Detailed methods information is lacking.

3) Fig.4F & 4G

- M1 should be diffused distrubted throughout cell upon single expression. It is unclear whether the cells were MG132-treated and there is no ZsGreen-only control.

- Right-most column: appears to be an aggregate, rather than a MTOC-associated aggresome.

Minor issues (not raised in previous review):

1) Figure 1E:

Does this figure indicate binding of TfR1 to viral components even without UV crosslinking? Please clarify the intended conclusion from Fig. 1E.

2) Figures 2A, 2D, S2C, and others:

Virus entry/infection was measured by the expression of intracellular M1 protein, yet the authors claim that M1 is degraded upon cell entry via TfR1. This method is problematic and confusing because differences in M1 levels might reflect differential degradation rather than differences in virus entry. We recommend quantifying virus entry/infection using nucleoprotein (NP), a viral protein unaffected by TfR1 expression according to the authors’ own data.

3) Figure 2C:

Clarify the exact hours post-infection when these cells were collected.

4) Figure S5D:

Please indicate which data correspond to the "IAV plasmid transfected HEK293T cells" mentioned in the main text.

5) Page 10 text and Figure S6C mismatch:

The statement, “Further truncations demonstrated that expression of the intracellular and transmembrane domain...” does not match the data shown in Fig. S6C. Please correct or clarify this discrepancy.

6) Figure 5C and 5D (TfR1 truncated mutants):

The subcellular localization of truncated TfR1 mutants (e.g., Δ1-63, ΔTM) needs to be shown. At least for mutants affecting M1 degradation capability, localization data are essential to confirm whether functional differences might be due to altered cellular localization. The M1 blot is missing.

7) Figure 6B (label clarification):

Please clarify what the term "cyto" means in Fig. 6B. Is this equivalent to the "intracellular domain" described in Fig. 5D?

PLOS authors have the option to publish the peer review history of their article (what does this mean? ). If published, this will include your full peer review and any attached files.

**Do you want your identity to be public for this peer review?** For information about this choice, including consent withdrawal, please see our Privacy Policy .

Reviewer #2: No

Reviewer #3: No

**Figure resubmission:**
---

## [Decision Letter · Decision Letter 2]

2 Sep 2025

Dear Prof. and Dr. Zhou,

We are pleased to inform you that your manuscript 'TfR1 facilitates influenza virus endocytosis and uncoating by interacting with NA and M1 via extracellular and intracellular domains' has been provisionally accepted for publication in PLOS Pathogens.

Best regards,

Maria João Amorim, Ph.D

Academic Editor

PLOS Pathogens

Matthias Schnell

Section Editor

PLOS Pathogens

Sumita Bhaduri-McIntosh

Editor-in-Chief

PLOS Pathogens

orcid.org/0000-0003-2946-9497

Michael Malim

Editor-in-Chief

PLOS Pathogens

orcid.org/0000-0002-7699-2064

Reviewer #3:

Reviewer Comments (if any, and for reference):

Reviewer's Responses to Questions

**Part I - Summary**

Reviewer #3: Overall, the authors have adequately addressed the reviewer's points by providing clarifications, additional data, and methodological improvements. The added content should enhance the clarity of the manuscript.

**Part II – Major Issues: Key Experiments Required for Acceptance**

Reviewer #3: (No Response)

**Part III – Minor Issues: Editorial and Data Presentation Modifications**

Reviewer #3: (No Response)

PLOS authors have the option to publish the peer review history of their article (what does this mean? ). If published, this will include your full peer review and any attached files.

**Do you want your identity to be public for this peer review?** For information about this choice, including consent withdrawal, please see our Privacy Policy .

Reviewer #3: No

---

## [Editor Report · Acceptance letter]

Dear Prof. and Dr. Zhou,

We are delighted to inform you that your manuscript, "TfR1 facilitates influenza virus endocytosis and uncoating by interacting with NA and M1 via extracellular and intracellular domains," has been formally accepted for publication in PLOS Pathogens.

Best regards,

Sumita Bhaduri-McIntosh

Editor-in-Chief

PLOS Pathogens

orcid.org/0000-0003-2946-9497

Michael Malim

Editor-in-Chief

PLOS Pathogens

orcid.org/0000-0002-7699-2064